# Average-case Acceleration for Bilinear Games and Normal Matrices

**Carles Domingo-Enrich**
Computer Science Department
Courant Institute of Mathematical Sciences
New York University
New York, NY 10012, USA
cd2754@nyu.edu

**Fabian Pedregosa**
Google Research
pedregosa@google.com

**Damien Scieur**
Samsung SAIT AI Lab & Mila
Montreal, Canada
damien.scieur@gmail.com

## Abstract

Advances in generative modeling and adversarial learning have given rise to renewed interest in smooth games. However, the absence of symmetry in the matrix of second derivatives poses challenges that are not present in the classical minimization framework. While a rich theory of average-case analysis has been developed for minimization problems, little is known in the context of smooth games. In this work we take a first step towards closing this gap by developing average-case optimal first-order methods for a subset of smooth games. We make the following three main contributions. First, we show that for zero-sum bilinear games the average-case optimal method is the optimal method for the minimization of the Hamiltonian. Second, we provide an explicit expression for the optimal method corresponding to normal matrices, potentially non-symmetric. Finally, we specialize it to matrices with eigenvalues located in a disk and show a provable speed-up compared to worst-case optimal algorithms. We illustrate our findings through numerical simulations with a varying degree of mismatch with our assumptions.

## 1 Introduction

The traditional analysis of optimization algorithms is a worst-case analysis (Nemirovski, 1995; Nesterov, 2004). This type of analysis provides a complexity bound for any input from a function class, no matter how unlikely. However, since hard-to-solve inputs might rarely occur in practice, the worst-case complexity bounds might not be representative of the observed running time.

A more representative analysis is given by the average-case complexity, averaging the algorithm's complexity over all possible inputs. This analysis is standard for analyzing, e.g., sorting (Knuth, 1997) and cryptography algorithms (Katz & Lindell, 2014). Recently, a line of work (Berthier et al., 2020; Pedregosa & Scieur, 2020; Lacotte & Pilanci, 2020; Paquette et al., 2020) focused on optimal methods for the optimization of quadratics, specified by a symmetric matrix. While worst-case analysis uses bounds on the matrix eigenvalues to yield upper and lower bounds on convergence, average-case analysis relies on the expected distribution of eigenvalues and provides algorithms with sharp optimal convergence rates. While the algorithms developed in this context have been shown to be efficient for minimization problems, these have not been extended to smooth games.

A different line of work considers algorithms for smooth games but studies *worst-case* optimal methods (Azizian et al., 2020). In this work, we combine average-case analysis with smooth games, and develop novel average-case optimal algorithms for finding the root of a linear system determined by a (potentially non-symmetric) normal matrix. We make the following **main contributions**:

1. Inspired by the problem of finding equilibria in smooth games, we develop average-case optimal algorithms for finding the root of a non-symmetric affine operator, both under a normality assumption (Thm. 4.1), and under the extra assumption that eigenvalues of the operator are supported in a disk (Thm. 4.2). The proposed method shows a polynomial speedup compared to the worst-case optimal method, verified by numerical simulations.

2. We make a novel connection between average-case optimal methods for optimization, and average-case optimal methods for bilinear games. In particular, we show that solving the Hamiltonian using an average-case optimal method is optimal (Theorem 3.1) for bilinear games. This result complements (Azizian et al., 2020), who proved that Polyak Heavy Ball algorithm on the Hamiltonian is asymptotically worst-case optimal for bilinear games.

## 2 Average-case analysis for normal matrices

In this paper we consider the following class of problems.

**Definition 1.** *Let $A \in \mathbb{R}^{d \times d}$ be a real matrix and $x^\star \in \mathbb{R}^d$ a vector. The non-symmetric (affine) operator (**NSO**) problem is defined as:*

$$\text{Find } x \ : \ F(x) \stackrel{def}{=} A(x - x^\star) = 0 \,. \tag{NSO}$$

This problem generalizes that of minimization of a convex quadratic function $f$, since we can cast the latter in this framework by setting the operator $F = \nabla f$. The set of solutions is an affine subspace that we will denote $\mathcal{X}^\star$. We will find convenient to consider the distance to this set, defined as

$$\text{dist}(x, \mathcal{X}^\star) \stackrel{def}{=} \min_{v \in \mathcal{X}^\star} \|x - v\|^2, \quad \text{with } \mathcal{X}^\star = \{x \in \mathbb{R}^d \,|\, A(x - x^\star) = 0\} \,. \tag{1}$$

In this paper we will develop *average-case* optimal methods. For this, we consider $A$ and $x^\star$ to be random vectors, and a random initialization $x_0$. This induces a probability distribution over NSO problems, and we seek to find methods that have an optimal *expected* suboptimality w.r.t. this distribution. Denoting $\mathbb{E}_{(A, x^\star, x_0)}$ the expectation over these random problems, we have that average-case optimal methods they verify the following property at each iteration $t$

$$\min_{x_t} \mathbb{E}_{(A, x^\star, x_0)} \text{dist}(x_t, \mathcal{X}^\star) \quad \text{s.t.} \ \ x_i \in x_0 + \text{span}(\{F(x_j)\}_{j=0}^{i-1}), \ \forall i \in [1:t]. \tag{2}$$

The last condition on $x_t$ stems from restricting the class of algorithms to first-order methods. The class of first-order methods encompasses many known schemes such as gradient descent with momentum, or full-matrix AdaGrad. However, methods such as Adam (Kingma & Ba, 2015) or diagonal AdaGrad (Duchi et al., 2011) are *not* in this class, as the diagonal re-scaling creates iterates $x_t$ outside the span of previous gradients. Although we will focus on the distance to the solution, the results can be extended to other convergence criteria such as $\|F(x_t)\|^2$.

Finally, note that the expectations in this paper are on the problem instance and *not* on the randomness of the algorithm.

### 2.1 Orthogonal residual polynomials and first-order methods

The analysis of first-order methods simplifies through the use of polynomials. This section provides the tools required to leverage this connection.

**Definition 2.** *A **residual polynomial** is a polynomial $P$ that satisfies $P(0) = 1$.*

**Proposition 2.1.** *(Hestenes et al., 1952) If the sequence $(x_t)_{t \in \mathbb{Z}_+}$ is generated by a first-order method, then there exist residual polynomials $P_t$, each one of degree at most $t$, verifying*

$$x_t - x^\star = P_t(A)(x_0 - x^\star). \tag{3}$$

As we will see, optimal average-case method are strongly related to orthogonal polynomials. We first define the inner product between polynomials, where we use $z^*$ for the complex conjugate of $z \in \mathbb{C}$.

**Definition 3.** *For $P, Q \in \mathbb{R}[X]$, we define the inner product $\langle \cdot, \cdot \rangle_\mu$ for a measure $\mu$ over $\mathbb{C}$ as*

$$\langle P, Q \rangle_\mu \overset{def}{=} \int_\mathbb{C} P(\lambda) Q(\lambda)^* \, \mathrm{d}\mu(\lambda) \,. \tag{4}$$

**Definition 4.** *A sequence of polynomials $\{P_i\}$ is **orthogonal** (resp. **orthonormal**) w.r.t. $\langle \cdot, \cdot \rangle_\mu$ if*

$$\langle P_i, P_i \rangle_\mu > 0 \ \ (\text{resp.} = 1); \qquad \langle P_i, P_j \rangle_\mu = 0 \ \ \text{if } i \neq j.$$

## 2.2 Expected Spectral Distribution

Following (Pedregosa & Scieur, 2020), we make the following assumption on the problem family.

**Assumption 1.** $\boldsymbol{x}_0 - \boldsymbol{x}^\star$ is independent of $\boldsymbol{A}$, and $\mathbb{E}_{(\boldsymbol{x}_0, \boldsymbol{x}^\star)}[(\boldsymbol{x}_0 - \boldsymbol{x}^\star)(\boldsymbol{x}_0 - \boldsymbol{x}^\star)^\top] = \frac{R^2}{d} \boldsymbol{I}_d$.

We will also require the following definitions to characterize difficulty of a problem class. Let $\{\lambda_1, \ldots, \lambda_d\}$ be the eigenvalues of a matrix $\boldsymbol{A} \in \mathbb{R}^{d \times d}$. We define the **empirical spectral distribution** of $\boldsymbol{A}$ as the probability measure

$$\hat{\mu}_{\boldsymbol{A}}(\lambda) \overset{def}{=} \tfrac{1}{d} \sum_{i=1}^d \delta_{\lambda_i}(\lambda) \,, \tag{5}$$

where $\delta_{\lambda_i}$ is the Dirac delta, a distribution equal to zero everywhere except at $\lambda_i$ and whose integral over the entire real line is equal to one. Note that with this definition, $\int_\mathcal{D} \mathrm{d}\hat{\mu}_{\boldsymbol{A}}(\lambda)$ corresponds to the proportion of eigenvalues in $\mathcal{D}$.

When $\boldsymbol{A}$ is a matrix-valued random variable, $\mu_{\boldsymbol{A}}$ is a measure-valued random variable. As such, we can define its **expected spectral distribution**

$$\mu_{\boldsymbol{A}} \overset{def}{=} \mathbb{E}_{\boldsymbol{A}}[\hat{\mu}_{\boldsymbol{A}}] \,, \tag{6}$$

which by the Riesz representation theorem is the measure that verifies $\int f \, \mathrm{d}\mu = \mathbb{E}_{\boldsymbol{A}}[\int f \, \mathrm{d}\mu_{\boldsymbol{A}}]$ for all measureable $f$. Surprisingly, the expected spectral distribution is the only required characteristic to design optimal algorithms in the average-case.

## 2.3 Expected error of first-order methods

In this section we provide an expression for the expected convergence in terms of the residual polynomial and the expected spectral distribution introduced in the previous section. To go further in the analysis, we have to assume that $\boldsymbol{A}$ is a normal matrix.

**Assumption 2.** *The (real) random matrix $\boldsymbol{A}$ is normal, that is, it verifies $\boldsymbol{A}\boldsymbol{A}^\top = \boldsymbol{A}^\top \boldsymbol{A}$.*

Normality is equivalent to $\boldsymbol{A}$ having the spectral decomposition $\boldsymbol{A} = \boldsymbol{U}\boldsymbol{\Lambda}\boldsymbol{U}^*$, where $\boldsymbol{U}$ is unitary, i.e., $\boldsymbol{U}^*\boldsymbol{U} = \boldsymbol{U}\boldsymbol{U}^* = \boldsymbol{I}$. We now have everything to write the expected error of a first-order algorithm applied to (NSO).

---

**Theorem 2.1.** *Consider the application of a first-order method associated to the sequence of polynomials $\{P_t\}$ (Proposition 2.1) on the problem (NSO). Let $\mu$ be the expected spectral distribution of $\boldsymbol{A}$. Under Assumptions 1 and 2, we have*

$$\mathbb{E}[\mathrm{dist}(\boldsymbol{x}_t, \mathcal{X}^\star)] = R^2 \int_{\mathbb{C} \setminus \{0\}} |P_t|^2 \, \mathrm{d}\mu \,, \tag{7}$$

---

Before designing optimal algorithms for certain specific distributions, we compare our setting with the average-case accelerating for minimization problems of Pedregosa & Scieur (2020), who proposed optimal *optimization* algorithms in the average-case.

## 2.4 Difficulties of First-Order Methods on Games and Related Work

This section compares our contribution with the existing framework of average-case optimal methods for quadratic minimization problems.

**Definition 5.** *Let $\boldsymbol{H} \in \mathbb{R}^{d \times d}$ be a random symmetric positive-definite matrix and $\boldsymbol{x}^\star \in \mathbb{R}^d$ a random vector. These elements determine the following **random quadratic minimization problem***

$$\min_{\boldsymbol{x} \in \mathbb{R}^d} \left\{ f(\boldsymbol{x}) \stackrel{def}{=} \frac{1}{2} (\boldsymbol{x} - \boldsymbol{x}^\star)^\top \boldsymbol{H} (\boldsymbol{x} - \boldsymbol{x}^\star) \right\} . \qquad \text{(OPT)}$$

As in our paper, Pedregosa & Scieur (2020) find deterministic optimal first-order algorithms in *expectation* w.r.t. the matrix $\boldsymbol{H}$, the solution $\boldsymbol{x}^\star$, and the initialization $\boldsymbol{x}_0$. Since they work with problem (OPT), their problem is equivalent to (NSO) with the matrix $\boldsymbol{A} = \boldsymbol{H}$. However, they have the *stronger* assumption that the matrix is *symmetric*, which implies being normal. The normality assumption is restrictive in the case of game theory, as they do not always naturally fit such applications. However, this set is expressive enough to consider interesting cases, such as bilinear games, and our experiments show that our findings are also consistent with non-normal matrices.

Using orthogonal residual polynomials and spectral distributions, they derive the explicit formula of the expected error. Their result is similar to Theorem 2.1, but the major difference is the domain of the integral, a real positive line in convex optimization, but a shape in the complex plane in our case. This shape plays a crucial role in the rate of converge of first-order algorithms, as depicted in the work of Azizian et al. (2020); Bollapragada et al. (2018).

In the case of optimization methods, they show that optimal schemes in the average-case follow a simple three-term recurrence arising from the three-term recurrence for residual orthogonal polynomials for the measure $\lambda \mu(\lambda)$. Indeed, by Theorem 2.1 the optimal method corresponds to the residual polynomials minimizing $\langle P, P \rangle_\mu$, and the following result holds:

**Theorem 2.2.** *(Fischer, 1996, §2.4) When $\mu$ is supported in the real line, the residual polynomial of degree $t$ minimizing $\langle P, P \rangle_\mu$ is given by the degree $t$ residual orthogonal polynomial w.r.t. $\lambda \mu(\lambda)$.*

However, the analogous result does not hold for general measures in $\mathbb{C}$, and hence our arguments will make use of the following Theorem 2.3 instead, which links the residual polynomial of degree at most $t$ that minimizes $\langle P, P \rangle_\mu$ to the sequence of orthonormal polynomials for $\mu$.

**Theorem 2.3.** *[Theorem 1.4 of Assche (1997)] Let $\mu$ be a positive Borel measure in the complex plane. The minimum of the integral $\int_\mathbb{C} |P(\lambda)|^2 \, \mathrm{d}\mu(\lambda)$ over residual polynomials $P$ of degree lower or equal than $t$ is uniquely attained by the polynomial*

$$P^\star(\lambda) = \frac{\sum_{k=0}^t \phi_k(\lambda) \phi_k(0)^*}{\sum_{k=0}^t |\phi_k(0)|^2}, \quad \text{with optimal value} \quad \int_\mathbb{C} |P^\star(\lambda)|^2 \, \mathrm{d}\mu(\lambda) = \frac{1}{\sum_{k=0}^t |\phi_k(0)|^2} , \quad (8)$$

*where $(\phi_k)_k$ is the orthonormal sequence of polynomials with respect to the inner product $\langle \cdot, \cdot \rangle_\mu$.*

In the next sections we consider cases where the optimal scheme is identifiable.

## 3 Average-case Optimal Methods for Bilinear Games

We consider the problem of finding a Nash equilibrium of the zero-sum minimax game given by

$$\min_{\boldsymbol{\theta}_1} \max_{\boldsymbol{\theta}_2} \ell(\boldsymbol{\theta}_1, \boldsymbol{\theta}_2) \stackrel{\text{def}}{=} (\boldsymbol{\theta}_1 - \boldsymbol{\theta}_1^\star)^\top \boldsymbol{M} (\boldsymbol{\theta}_2 - \boldsymbol{\theta}_2^\star) . \qquad (9)$$

Let $\boldsymbol{\theta}_1, \boldsymbol{\theta}_1^\star \in \mathbb{R}^{d_1}, \boldsymbol{\theta}_2, \boldsymbol{\theta}_2^\star \in \mathbb{R}^{d_2}, \boldsymbol{M} \in \mathbb{R}^{d_1 \times d_2}$ and $d \stackrel{\text{def}}{=} d_1 + d_2$. The vector field of the game (Balduzzi et al., 2018) is defined as $F(\boldsymbol{x}) = \boldsymbol{A}(\boldsymbol{x} - \boldsymbol{x}^\star)$, where

$$F(\boldsymbol{\theta}_1, \boldsymbol{\theta}_2) = \begin{bmatrix} \nabla_{\boldsymbol{\theta}_1} \ell(\boldsymbol{\theta}_1, \boldsymbol{\theta}_2) \\ -\nabla_{\boldsymbol{\theta}_2} \ell(\boldsymbol{\theta}_1, \boldsymbol{\theta}_2) \end{bmatrix} = \underbrace{\begin{bmatrix} 0 & \boldsymbol{M} \\ -\boldsymbol{M}^\top & 0 \end{bmatrix}}_{=\boldsymbol{A}} \left( \underbrace{\begin{bmatrix} \boldsymbol{\theta}_1 \\ \boldsymbol{\theta}_2 \end{bmatrix}}_{=\boldsymbol{x}} - \underbrace{\begin{bmatrix} \boldsymbol{\theta}_1^\star \\ \boldsymbol{\theta}_2^\star \end{bmatrix}}_{=\boldsymbol{x}^\star} \right) = \boldsymbol{A}(\boldsymbol{x} - \boldsymbol{x}^\star) . \qquad (10)$$

As before, $\mathcal{X}^\star$ denotes the set of points $\boldsymbol{x}$ such that $F(\boldsymbol{x}) = 0$, which is equivalent to the set of Nash equilibrium. If $\boldsymbol{M}$ is sampled independently from $\boldsymbol{x}_0, \boldsymbol{x}^\star$ and $\boldsymbol{x}_0 - \boldsymbol{x}^\star$ has covariance $\frac{R^2}{d} \boldsymbol{I}_d$, Assumption 1 is fulfilled. Since $\boldsymbol{A}$ is skew-symmetric, it is in particular normal and Assumption 2 is also satisfied.

We now show that the optimal average-case algorithm to solve bilinear problems is Hamiltonian gradient descent with momentum, described below in its general form. Contrary to the methods in Azizian et al. (2020), the method we propose is *anytime* (and not only asymptotically) average-case optimal.

---

**Optimal average-case algorithm for bilinear games.**

**Initialization.** $\boldsymbol{x}_{-1} = \boldsymbol{x}_0 = \left(\boldsymbol{\theta}_{1,0}, \ \boldsymbol{\theta}_{2,0}\right)$, sequence $\{h_t, m_t\}$ given by Theorem 3.1.
**Main loop.** For $t \geq 0$,

$$
\begin{aligned}
\boldsymbol{g}_t &= F(\boldsymbol{x}_t - F(\boldsymbol{x}_t)) - F(\boldsymbol{x}_t) && \left(= \tfrac{1}{2}\nabla \|F(\boldsymbol{x}_t)\|^2 \ \text{ by (12)}\right) \\
\boldsymbol{x}_{t+1} &= \boldsymbol{x}_t - h_{t+1}\boldsymbol{g}_t + m_{t+1}(\boldsymbol{x}_{t-1} - \boldsymbol{x}_t)
\end{aligned}
\tag{11}
$$

---

The quantity $\frac{1}{2}\|F(\boldsymbol{x})\|^2$ is commonly known as the Hamiltonian of the game (Balduzzi et al., 2018), hence the name *Hamiltonian gradient descent*. Indeed, $\boldsymbol{g}_t = \nabla\left(\frac{1}{2}\|F(\boldsymbol{x})\|^2\right)$ when $F$ is affine:

$$
\begin{aligned}
F(\boldsymbol{x} - F(\boldsymbol{x})) - F(\boldsymbol{x}) &= \boldsymbol{A}(\boldsymbol{x} - \boldsymbol{A}(\boldsymbol{x} - \boldsymbol{x}^\star) - \boldsymbol{x}^\star) - \boldsymbol{A}(\boldsymbol{x} - \boldsymbol{x}^\star) = -\boldsymbol{A}(\boldsymbol{A}(\boldsymbol{x} - \boldsymbol{x}^\star)) \\
&= \boldsymbol{A}^\top(\boldsymbol{A}(\boldsymbol{x} - \boldsymbol{x}^\star)) = \nabla\left(\frac{1}{2}\|\boldsymbol{A}(\boldsymbol{x} - \boldsymbol{x}^\star)\|^2\right) = \nabla\left(\frac{1}{2}\|F(\boldsymbol{x})\|^2\right).
\end{aligned}
\tag{12}
$$

The following theorem shows that (11) is indeeed the optimal average-case method associated to the minimization problem $\min_{\boldsymbol{x}} \left(\frac{1}{2}\|F(\boldsymbol{x})\|^2\right)$, as the following theorem shows.

---

**Theorem 3.1.** *Suppose that Assumption 1 holds and that the expected spectral distribution of $\boldsymbol{M}\boldsymbol{M}^\top$ is absolutely continuous with respect to the Lebesgue measure. Then, the method (11) is average-case optimal for bilinear games when $h_t$, $m_t$ are chosen to be the coefficients of the average-case optimal minimization of $\frac{1}{2}\|F(\boldsymbol{x})\|^2$.*

---

**How to find optimal coefficients?** Since $\frac{1}{2}\|F(\boldsymbol{x})\|^2$ is a quadratic problem, the coefficients $\{h_t, m_t\}$ can be found using the average-case framework for quadratic minimization problems of (Pedregosa & Scieur, 2020, Theorem 3.1).

*Proof sketch.* When computing the optimal polynomial $\boldsymbol{x}_t = P_t(\boldsymbol{A})(\boldsymbol{x}_0 - \boldsymbol{x}^\star)$, we have that the residual orthogonal polynomial $P_t$ behaves differently if $t$ is even or odd.

- **Case 1: $t$ is even.** In this case, we observe that the polynomial $P_t(\boldsymbol{A})$ can be expressed as $Q_{t/2}(-\boldsymbol{A}^2)$, where $(Q_t)_{t\geq 0}$ is the sequence of orthogonal polynomials w.r.t. the expected spectral density of $-\boldsymbol{A}^2$, whose eigenvalues are real and positive. This gives the recursion in (11).

- **Case 2: $t$ is odd.** There is no residual orthogonal polynomial of degree $t$ for $t$ odd. Instead, odd iterations do correspond to the intermediate computation of $\boldsymbol{g}_t$ in (11), but not to an actual iterate.

## 3.1 Particular case: $\boldsymbol{M}$ with i.i.d. components

We now show the optimal method when the entries of $\boldsymbol{M}$ are i.i.d. sampled. For simplicity, we order the players such that $d_1 \leq d_2$.

**Assumption 3.** *Assume that each component of $\boldsymbol{M}$ is sampled iid from a distribution of mean 0 and variance $\sigma^2$, and we take $d_1, d_2 \to \infty$ with $\frac{d_1}{d_2} \to r < 1$.*

In such case, the spectral distribution of $\frac{1}{d_2}\boldsymbol{M}\boldsymbol{M}^\top$ tends to the Marchenko-Pastur law, supported in $[\ell, L]$ and with density:

$$
\rho_{MP}(\lambda) \stackrel{\text{def}}{=} \frac{\sqrt{(L-\lambda)(\lambda-\ell)}}{2\pi\sigma^2 r\lambda}, \quad \text{where } L \stackrel{\text{def}}{=} \sigma^2(1+\sqrt{r})^2, \ell \stackrel{\text{def}}{=} \sigma^2(1-\sqrt{r})^2.
\tag{13}
$$

**Proposition 3.1.** *When $M$ satisfies Assumption 3, the optimal parameter of scheme (11) are*

$$h_t = -\frac{\delta_t}{\sigma^2\sqrt{r}}, \quad m_t = 1 + \rho\delta_t, \quad \text{where} \ \ \rho = \frac{1+r}{\sqrt{r}}, \ \ \delta_t = (-\rho - \delta_{t-1})^{-1}, \ \ \delta_0 = 0. \quad (14)$$

*Proof.* By Theorem 3.1, the problem reduces to finding the optimal average-case algorithm for the problem $\min_{\boldsymbol{x}} \frac{1}{2}\|F(\boldsymbol{x})\|^2$. Since the expected spectral distribution of $\frac{1}{d_2}\boldsymbol{M}\boldsymbol{M}^\top$ is the Marchenko-Pastur law, we can use the optimal algorithm from (Pedregosa & Scieur, 2020, Section 5). $\square$

## 4 GENERAL AVERAGE-CASE OPTIMAL METHOD FOR NORMAL OPERATORS

In this section we derive general average-case optimal first-order methods for normal operators. First, we need to assume the existence of a three-term recurrence for residual orthogonal polynomials (Assumption 4). As mentioned in subsection 2.4, for general measures in the complex plane, the existence of a three-term recurrence of orthogonal polynomials is not ensured. In Proposition B.3 in Appendix B we give a sufficient condition for its existence, and in the next subsection we will show specific examples where the residual orthogonal polynomials satisfy the three-term recurrence.

**Assumption 4** (Simplifying assumption). *The sequence of residual polynomials $\{\psi_t\}_{t\geq 0}$ orthogonal w.r.t. the measure $\mu$, defined on the complex plane, admits the three-term recurrence*

$$\psi_{-1} = 0, \quad \psi_0 = 1, \quad \psi_t(\lambda) = (a_t + b_t\lambda)\psi_{t-1}(\lambda) + (1 - a_t)\psi_{t-2}(\lambda). \quad (15)$$

Under Assumption 4, Theorem 4.1 shows that the optimal algorithm can also be written as an average of iterates following a simple three-terms recurrence.

---

**Theorem 4.1.** *Under Assumption 4 and the assumptions of Theorem 2.1, the following algorithm is optimal in the average case, with $\boldsymbol{y}_{-1} = \boldsymbol{y}_0 = \boldsymbol{x}_0$:*

$$\boldsymbol{y}_t = a_t\boldsymbol{y}_{t-1} + (1 - a_t)\boldsymbol{y}_{t-2} + b_tF(\boldsymbol{y}_{t-1})$$

$$\boldsymbol{x}_t = \frac{B_t}{B_t + \beta_t}\boldsymbol{x}_{t-1} + \frac{\beta_t}{B_t + \beta_t}\boldsymbol{y}_t, \quad \beta_t = \phi_t^2(0), \quad B_t = B_{t-1} + \beta_{t-1}, \quad B_0 = 0. \quad (16)$$

*where $(\phi_k(0))_{k\geq 0}$ can be computed using the three-term recurrence (upon normalization). Moreover, $\mathbb{E}_{(\boldsymbol{A},\boldsymbol{x}^\star,\boldsymbol{x}_0)} \text{dist}(\boldsymbol{x}_t, \mathcal{X}^\star)$ converges to zero at rate $1/B_t$.*

---

**Remark.** Notice that it is not immediate that (16) fulfills the definition of first-order algorithms stated in (2), as $\boldsymbol{y}_t$ is clearly a first-order method but $\boldsymbol{x}_t$ is an average of the iterates $\boldsymbol{y}_t$. Using that $F$ is an affine function we see that $\boldsymbol{x}_t$ indeed fulfills (2).

**Remark.** Assumption 4 is needed for the sequence $(\boldsymbol{y}_t)_{t\geq 0}$ to be computable using a three-term recurrence. However, for some distribution, the associated sequence of orthogonal polynomials may admit another recurrence that may not satisfy Assumption 4.

### 4.1 CIRCULAR SPECTRAL DISTRIBUTIONS

In random matrix theory, the circular law states that if $\boldsymbol{A}$ is an $n \times n$ matrix with i.i.d. entries of mean $C$ and variance $R^2/n$, as $n \to \infty$ the spectral distribution of $\boldsymbol{A}$ tends to the uniform distribution on $D_{C,R}$. In this subsection we apply Theorem 4.1 to a class of spectral distributions specified by Assumption 5, which includes the uniform distribution on $D_{C,R}$. Even though the random matrices with i.i.d entries are not normal, we will see in section 6 that the empirical results for such matrices are consistent with our theoretical results under the normality assumption.

**Assumption 5.** *Assume that the expected spectral distribution $\mu_{\boldsymbol{A}}$ is supported in the complex plane on the disk $D_{C,R}$ of center $C \in \mathbb{R}, C > 0$ and radius $R < C$. Moreover, assume that the spectral density is circularly symmetric, i.e. there exists a probability measure $\mu_R$ supported on $[0, R]$ such for all $f$ measurable and $r \in [0, R]$, $\mathrm{d}\mu_{\boldsymbol{A}}(C + re^{i\theta}) = \frac{1}{2\pi} \mathrm{d}\theta \, \mathrm{d}\mu_R(r)$.*

**Proposition 4.1.** *If $\mu$ satisfies Assumption 5, the sequence of orthonormal polynomials is $(\phi_t)_{t\geq 0}$,*

$$\phi_t(\lambda) = \frac{(\lambda - C)^t}{K_{t,R}}, \quad \text{where} \ K_{t,R} = \sqrt{\int_0^R r^{2t} \, \mathrm{d}\mu_R(r)}. \quad (17)$$

**Example.** The uniform distribution in $D_{C,R}$ is to $\mathrm{d}\mu_R = \frac{2r}{R^2}\,\mathrm{d}r$, and $K_{t,R} = R^t/\sqrt{t+1}$.

From Proposition 4.1, the sequence of residual polynomials is given by $\phi_t(\lambda)/\phi_t(0) = \left(1 - \frac{\lambda}{C}\right)^t$, which implies that Assumption 4 is fulfilled with $a_t = 1, b_t = -\frac{1}{C}$. Thus, by Theorem 4.1 we have

---

**Theorem 4.2.** *Given an initialization $\boldsymbol{x}_0(\boldsymbol{y}_0 = \boldsymbol{x}_0)$, if Assumption 5 is fulfilled with $R < C$ and the assumptions of Theorem 2.1 hold, then the average-case optimal first-order method is*

$$\boldsymbol{y}_t = \boldsymbol{y}_{t-1} - \frac{1}{C}F(\boldsymbol{y}_{t-1}), \quad \beta_t = C^{2t}/K_{t,R}^2, \quad B_t = B_{t-1} + \beta_{t-1},$$

$$\boldsymbol{x}_t = \frac{B_t}{B_t + \beta_t}\boldsymbol{x}_{t-1} + \frac{\beta_t}{B_t + \beta_t}\boldsymbol{y}_t. \tag{18}$$

*Moreover, $\mathbb{E}_{(\boldsymbol{A},\boldsymbol{x}^\star,\boldsymbol{x}_0)}\,\mathrm{dist}(\boldsymbol{x}_t, \mathcal{X}^\star)$ converges to zero at rate $1/B_t$.*

---

We now compare Theorem 4.2 with worst-case methods studied in Azizian et al. (2020). They give a worst-case convergence lower bound of $(R/C)^{2t}$ on the quantity $\mathrm{dist}(\boldsymbol{z}_t, \mathcal{X}^\star)$ for first-order methods $(\boldsymbol{z}_t)_{t\geq 0}$ on matrices with eigenvalues in the disk $D_{C,R}$. By the classical analysis of first-order methods, this rate is achievable by gradient descent with stepsize $1/C$, i.e. the iterates $\boldsymbol{y}_t$ defined in (18). However, by equation (79) in Proposition D.3 we have that under slight additional assumptions (those of Proposition 5.2), $\lim_{t\to\infty} \mathbb{E}\left[\mathrm{dist}(\boldsymbol{x}_t, \mathcal{X}^\star)\right]/\mathbb{E}\left[\mathrm{dist}(\boldsymbol{y}_t, \mathcal{X}^\star)\right] = 1 - \frac{R^2}{C^2}$ holds. That is, the average-case optimal algorithm outperforms gradient descent by a constant factor depending on the conditioning $R/C$.

## 5  ASYMPTOTIC BEHAVIOR

The recurrence coefficients of the average-case optimal method typically converges to limiting values when $t \to \infty$, which gives an "average-case asymptotically optimal first-order method" with constant coefficients. For the case of symmetric operators with spectrum in $[\ell, L]$, Scieur & Pedregosa (2020) show that under mild conditions, the asymptotically optimal algorithm is the Polyak momentum method with coefficients depending only on $\ell$ and $L$. For bilinear games, since the average-case optimal algorithm is the average-case optimal algorithm of an optimization algorithm, we can make use of their framework to obtain the asymptotic algorithm (see Theorem 3 of Scieur & Pedregosa (2020)).

**Proposition 5.1.** *Assume that the expected spectral density $\mu_{\boldsymbol{M}\boldsymbol{M}^\top}$ of $\boldsymbol{M}\boldsymbol{M}^\top$ is supported in $[\ell, L]$ for $0 < \ell < L$, and strictly positive in this interval. Then, the asymptotically optimal algorithm for bilinear games is the following version of Polyak momentum:*

$$\boldsymbol{g}_t = F(\boldsymbol{x}_t - F(\boldsymbol{x}_t)) - F(\boldsymbol{x}_t)$$

$$\boldsymbol{x}_{t+1} = \boldsymbol{x}_t + \left(\frac{\sqrt{L}-\sqrt{\ell}}{\sqrt{L}+\sqrt{\ell}}\right)^2 (\boldsymbol{x}_{t-1} - \boldsymbol{x}_t) - \left(\frac{2}{\sqrt{L}+\sqrt{\ell}}\right)^2 \boldsymbol{g}_t \tag{19}$$

Notice that the algorithm in (19) is the worst-case optimal algorithm from Proposition 4 of Azizian et al. (2020). For the case of circularly symmetric spectral densities with support on disks, we can also compute the asymptotically optimal algorithm.

**Proposition 5.2.** *Suppose that the assumptions of Theorem 4.2 hold with $\mu_R \in \mathcal{P}([0, R])$ fulfilling $\mu_R([r, R]) = \Omega((R - r)^\kappa)$ for $r$ in $[r_0, R]$ for some $r_0 \in [0, R)$ and for some $\kappa \in \mathbb{Z}$. Then, the average-case asymptotically optimal algorithm is, with $\boldsymbol{y}_0 = \boldsymbol{x}_0$:*

$$\boldsymbol{y}_t = \boldsymbol{y}_{t-1} - \frac{1}{C}F(\boldsymbol{y}_{t-1}),$$

$$\boldsymbol{x}_t = \left(\frac{R}{C}\right)^2 \boldsymbol{x}_{t-1} + \left(1 - \left(\frac{R}{C}\right)^2\right) \boldsymbol{y}_t. \tag{20}$$

*Moreover, the convergence rate for this algorithm is asymptotically the same one as for the optimal algorithm in Theorem 4.2. Namely, $\lim_{t\to\infty} \mathbb{E}\left[\mathrm{dist}(\boldsymbol{x}_t, \mathcal{X}^\star)\right] B_t = 1$.*

The condition on $\mu_R$ simply rules out cases in which the spectral density has exponentially small mass around 1. It is remarkable that in algorithm (20) the averaging coefficients can be expressed so simply in terms of the quantity $R/C$. Notice also that while the convergence rate of the algorithm

is slower than the convergence rate for the optimal algorithm by definition, both rates match in the limit, meaning that the asymptotically optimal algorithm also outperforms gradient descent by a constant factor $1 - \frac{R^2}{C^2}$ in the limit $t \to \infty$.

## 6  EXPERIMENTS

We compare some of the proposed methods on settings with varying degrees of mismatch with our assumptions.

**Bilinear Games.**    We consider min-max bilinear problems of the form (10), where the entries of $M$ are generated i.i.d. from a standard Gaussian distribution. We vary the ratio $r = d/n$ parameter for $d = 1000$ and compare the average-case optimal method of Theorems 3.1 and 5.1, the asymptotic worst-case optimal method of (Azizian et al., 2020) and extragradient (Korpelevich, 1976). In all cases, we use the convergence-rate optimal step-size assuming knowledge of the edges of the spectral distribution.

The spectral density for these problems is displayed in the first row of Figure 1 and the benchmark results on the second row. Average-case optimal methods always outperform other methods, and the largest gain is in the ill-conditioned regime ($r \approx 1$).

**Circular Distribution.**    For our second experiment we choose $A$ as a matrix with iid Gaussian random entries, therefore the support of the distribution of its eigenvalue is a disk. Note that $A$ does not satisfy the normality assumption of Assumption 2. Figure 1 (third row) compares the average-case optimal methods from Theorems 4.2 and 5.2 on two datasets with different levels of conditioning. Note that the methods converge despite the violation of Assumption 2, suggesting a broader applicability than the one proven in this paper. We leave this investigation for future work.

## 7  DISCUSSION AND FUTURE RESEARCH DIRECTIONS

In this paper, we presented a general framework for the design of optimal algorithms in the average-case for affine operators $F$, whose underlying matrix is possibly non-symmetric. However, our approach presents some limitations, the major one being the restriction to normal matrices. Fortunately, the numerical experiments above suggests that this assumption can be relaxed. Developing a theory without that assumption is left for future work. Another avenue for future work is to analyze the nonlinear-case in which the non-symmetric operator $A$ is non-linear, as well as the case in which it is accessed through a stochastic estimator (as done by (Loizou et al., 2020) for the worst-case analysis).

ACKNOWLEDGEMENTS

C. Domingo-Enrich has been partially funded by "la Caixa" Foundation (ID 100010434), under agreement LCF/BQ/AA18/11680094, and partially funded by the NYU Computer Science Department.

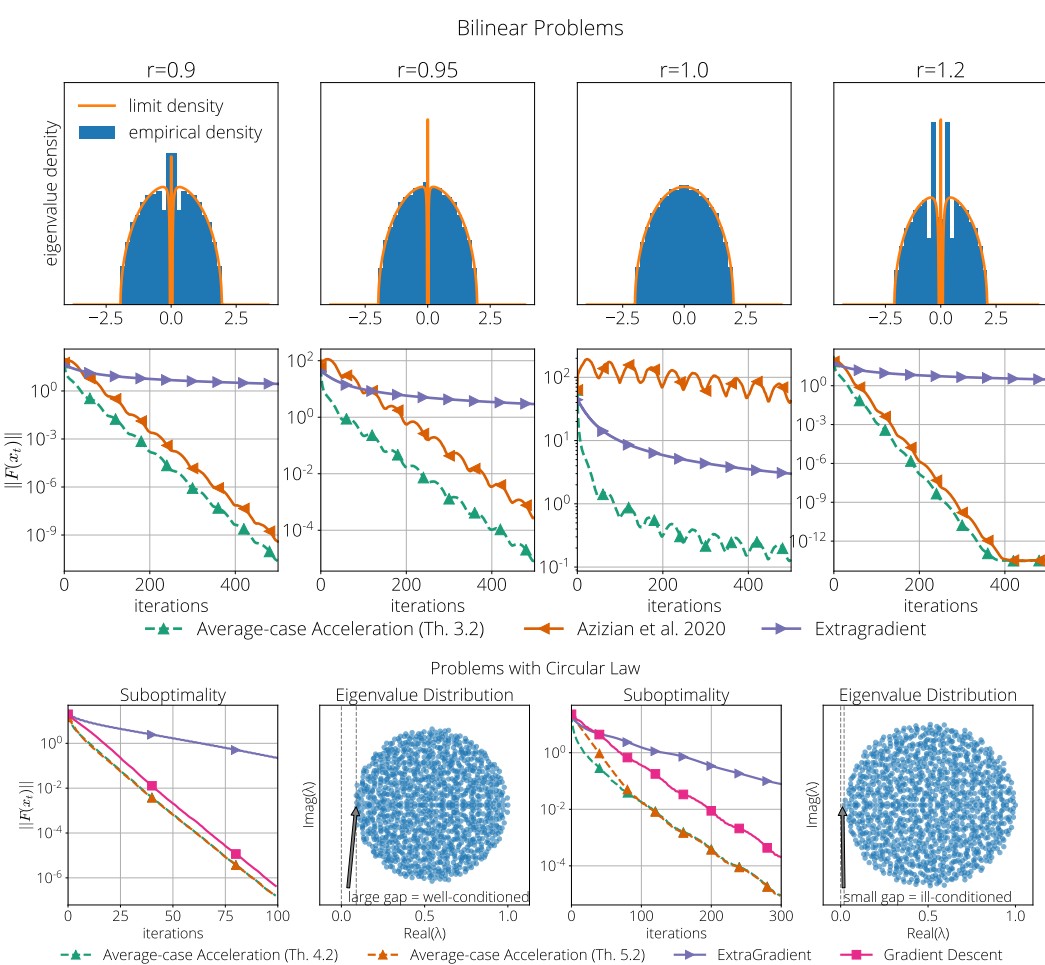

Figure 1: **Benchmarks and spectral density for different games.** *Top row*: spectral density associated with bilinear games for varying values of the ratio parameter $r = n/d$ (the x-axis represents the imaginary line). *Second row*: Benchmarks. Average-case optimal methods always outperform other methods, and the largest gain is in the ill-conditioned regime ($r \approx 1$). *Third row*. Benchmarks (columns 1 and 3) and eigenvalue distribution of a design matrix generated with iid entries for two different degrees of conditioning. Depite the normality assumption not being satisfied, we still observe an improvement of average-case optimal methods vs worst-case optimal ones.

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

# A    PROOF OF THEOREM 2.1

## A.1    PRELIMINARIES

Before proving Theorem 2.1, we quickly analyze the distance function (1), recalled below,

$$\text{dist}(\boldsymbol{x}, \mathcal{X}^\star) \stackrel{\text{def}}{=} \min_{\boldsymbol{v} \in \mathcal{X}^\star} \|\boldsymbol{x} - \boldsymbol{v}\|^2.$$

The definition of the distance function is not practical for the theoretical analysis. Fortunately, it is possible to find a simple expression that uses the orthogonal projection matrix $\Pi$ to the kernel $\text{Ker}(\boldsymbol{A})$. Since $\Pi$ is an orthogonal projection matrix to the kernel of a linear transformation, it satisfies

$$\Pi = \Pi^T, \quad \Pi^2 = \Pi, \quad \text{and} \quad \boldsymbol{A}\Pi = 0. \tag{21}$$

The normality assumption on $\boldsymbol{A}$ implies also that

$$\Pi \boldsymbol{A} = 0. \tag{22}$$

Indeed, the spectral decomposition of $\boldsymbol{A}$ is

$$\boldsymbol{A} = [\boldsymbol{U}_1 | \boldsymbol{U}_2] \begin{bmatrix} \boldsymbol{\Lambda} & 0 \\ 0 & 0 \end{bmatrix} [\boldsymbol{U}_1 | \boldsymbol{U}_2]^*,$$

and then $\Pi = \boldsymbol{U}_2 \boldsymbol{U}_2^*$. The next proposition uses $\Pi$ to derive the explicit solution of the (1).

**Proposition A.1.** *We have that*

$$\text{dist}(\boldsymbol{y}, \mathcal{X}^\star) = \|(\boldsymbol{I} - \Pi)(\boldsymbol{y} - \boldsymbol{x}^\star)\|^2 \quad \forall \boldsymbol{x}^\star \in \mathcal{X}^\star.$$

*Proof.* We first parametrize the set of solution $\mathcal{X}^\star$. By definition we have

$$\mathcal{X}^\star = \{\boldsymbol{x} : \boldsymbol{A}(\boldsymbol{x} - \boldsymbol{x}^\star) = 0\}.$$

Which can be written in terms of the kernel of $\boldsymbol{A}$ as

$$\mathcal{X}^\star = \{\boldsymbol{x}^\star + \Pi \boldsymbol{w} : \boldsymbol{w} \in \mathbb{R}^d\}.$$

From this, we can rewrite the distance function (1) as

$$\text{dist}(\boldsymbol{y}, \mathcal{X}^\star) = \min_{\boldsymbol{w} \in \mathbb{R}^d} \|\boldsymbol{y} - (\boldsymbol{x}^\star + \Pi \boldsymbol{w})\|^2.$$

The minimum can be attained at different points, but in particular at $\boldsymbol{w} = -(\boldsymbol{y} - \boldsymbol{x}^\star)$, which proves the statement. $\quad\square$

We now simplifies further the result of the previous proposition in the case where $\boldsymbol{x}_t$ is generated by a first order method.

**Proposition A.2.** *For every iterate $\boldsymbol{x}_t$ of a first-order methods, i.e., $\boldsymbol{x}_t$ satisfies*

$$\boldsymbol{x}_t - \boldsymbol{x}^\star = P_t(\boldsymbol{A})(\boldsymbol{x}_0 - \boldsymbol{x}^\star), \quad \deg(P_t) \leq t, \quad P(0) = \boldsymbol{I},$$

*we have that*

$$\text{dist}(\boldsymbol{x}_t, \mathcal{X}^\star) = \|\boldsymbol{x}_t - \boldsymbol{x}^\star\|^2 - \|\Pi(\boldsymbol{x}_0 - \boldsymbol{x}^\star)\|^2.$$

*Proof.* We start with the result of Proposition A.1,

$$\text{dist}(\boldsymbol{x}_t, \mathcal{X}^\star) = \|(\boldsymbol{I} - \Pi)(\boldsymbol{x}_t - \boldsymbol{x}^\star)\|^2.$$

The norm can be split into

$$\|(\boldsymbol{I} - \Pi)(\boldsymbol{x}_t - \boldsymbol{x}^\star)\|^2 = \|\boldsymbol{x}_t - \boldsymbol{x}^\star\|^2 + \|\underbrace{\Pi^2}_{=\Pi \text{ by } (21)}(\boldsymbol{x}_t - \boldsymbol{x}^\star)\|^2 - 2\|\Pi(\boldsymbol{x}_t - \boldsymbol{x}^\star)\|^2$$

$$= \|\boldsymbol{x}_t - \boldsymbol{x}^\star\|^2 - \|\Pi(\boldsymbol{x}_t - \boldsymbol{x}^\star)\|^2.$$

Since $\boldsymbol{x}_t$ is generated by a first order method, we have

$$\boldsymbol{x}_t - \boldsymbol{x}^\star = P_t(\boldsymbol{A})(\boldsymbol{x}_0 - \boldsymbol{x}^\star), \quad P_t(0) = 1.$$

Since $P(0) = 1$, the polynomial can be factorized as $P(\boldsymbol{A}) = \boldsymbol{I} + \boldsymbol{A}\boldsymbol{Q}_{t-1}(\boldsymbol{A})$, $\boldsymbol{Q}_{t-1}$ being a polynomial of degree $t - 1$. Therefore, $\|\Pi(\boldsymbol{x}_t - \boldsymbol{x}^\star)\|^2$ reads

$$
\begin{aligned}
\|\Pi(\boldsymbol{x}_t - \boldsymbol{x}^\star)\|^2 &= \|\Pi\left(\boldsymbol{I} + \boldsymbol{A}\boldsymbol{Q}_{t-1}(\boldsymbol{A})\right)(\boldsymbol{x}_0 - \boldsymbol{x}^\star)\|^2 \\
&= \|\Pi(\boldsymbol{x}_0 - \boldsymbol{x}^\star) + \underbrace{\Pi\boldsymbol{A}}_{=0 \text{ by } (22)} \boldsymbol{Q}_{t-1}(\boldsymbol{A})(\boldsymbol{x}_0 - \boldsymbol{x}^\star)\|^2 \\
&= \|\Pi(\boldsymbol{x}_0 - \boldsymbol{x}^\star)\|^2,
\end{aligned}
$$

which prove the statement. $\qquad\square$

### A.2 PROOF OF THE THEOREM

We are now ready to prove the main result.

**Theorem 2.1.** *Consider the application of a first-order method associated to the sequence of polynomials $\{P_t\}$ (Proposition 2.1) on the problem (NSO). Let $\mu$ be the expected spectral distribution of $\boldsymbol{A}$. Under Assumptions 1 and 2, we have*

$$
\mathbb{E}[\mathrm{dist}(\boldsymbol{x}_t, \mathcal{X}^\star)] = R^2 \int_{\mathbb{C}\backslash\{0\}} |P_t|^2 \, \mathrm{d}\mu, \tag{7}
$$

*Proof.* We start with the result of Proposition A.2,

$$
\mathrm{dist}(\boldsymbol{x}_t, \mathcal{X}^\star) = \|\boldsymbol{x}_t - \boldsymbol{x}^\star\|^2 - \|\Pi(\boldsymbol{x}_0 - \boldsymbol{x}^\star)\|^2.
$$

We now write the expectation of the distance function,

$$
\begin{aligned}
\mathbb{E}[\mathrm{dist}(\boldsymbol{x}_t, \mathcal{X}^\star)] &= \mathbb{E}\left[\|\boldsymbol{x}_t - \boldsymbol{x}^\star\|^2 - \|\Pi(\boldsymbol{x}_0 - \boldsymbol{x}^\star)\|^2\right] \\
&= \mathbb{E}\left[\|P_t(\boldsymbol{A})(\boldsymbol{x}_0 - \boldsymbol{x}^\star)\|^2 - \|\Pi(\boldsymbol{x}_0 - \boldsymbol{x}^\star)\|^2\right] \\
&= \mathbb{E}\left[\mathrm{tr}\, P_t(\boldsymbol{A})P_t(\boldsymbol{A})^T (\boldsymbol{x}_0 - \boldsymbol{x}^\star)(\boldsymbol{x}_0 - \boldsymbol{x}^\star)^T - \mathrm{tr}\,\Pi^2(\boldsymbol{x}_0 - \boldsymbol{x}^\star)(\boldsymbol{x}_0 - \boldsymbol{x}^\star)^T\right] \\
&= \mathbb{E}_A\left[\mathrm{tr}\, P_t(\boldsymbol{A})P_t(\boldsymbol{A})^T\mathbb{E}\left[(\boldsymbol{x}_0 - \boldsymbol{x}^\star)(\boldsymbol{x}_0 - \boldsymbol{x}^\star)^T |\boldsymbol{A}\right] - \mathrm{tr}\,\Pi\mathbb{E}\left[(\boldsymbol{x}_0 - \boldsymbol{x}^\star)(\boldsymbol{x}_0 - \boldsymbol{x}^\star)^T|\boldsymbol{A}\right]\right] \\
&= R\mathbb{E}_A\left[\mathrm{tr}\, P_t(\boldsymbol{A})P_t(\boldsymbol{A})^T - \mathrm{tr}\,\Pi\right] \\
&= R\mathbb{E}\left[\sum_{i=1}^d |P(\lambda_i)|^2 - \mathrm{tr}\,\Pi\right] \\
&= R\mathbb{E}\left[\int_{\mathbb{C}\backslash\{0\}} |P(\lambda)|^2\delta_{\lambda_i}(\lambda) + |P(0)|^2 \cdot [\# \text{ zero eigenvalues}] - \mathrm{tr}\,\Pi\right]
\end{aligned}
$$

However, $|P(0)|^2 = 1$ and $\mathrm{tr}\,\Pi$ corresponds to the number of zero eigenvalues of $\boldsymbol{A}$, therefore,

$$
E[\mathrm{dist}(\boldsymbol{x}_t, \mathcal{X}^\star)] = R\mathbb{E}\left[\int_{\mathbb{C}\backslash\{0\}} |P(\lambda)|^2\delta_{\lambda_i}(\lambda)\right] = R\int_{\mathbb{C}\backslash\{0\}} P(\lambda)\mu(\lambda).
$$

$\qquad\square$

## B PROOFS OF THEOREM 3.1 AND PROPOSITION 3.1

**Proposition B.1.** *[Block determinant formula] If $A, B, C, \boldsymbol{D}$ are (not necessarily square) matrices,*

$$
det \begin{bmatrix} \boldsymbol{A} & \boldsymbol{B} \\ \boldsymbol{C} & \boldsymbol{D} \end{bmatrix} = det(\boldsymbol{D})det(\boldsymbol{A} - \boldsymbol{B}\boldsymbol{D}^{-1}\boldsymbol{C}), \tag{23}
$$

*if D is invertible.*

**Definition 6** (Pushforward of a measure). *Recall that the pushforward $f_*\mu$ of a measure $\mu$ by a function $f$ is defined as the measure such that for all measurable $g$,*

$$\int g(\lambda)\,\mathrm{d}(f_*\mu)(\lambda) = \int g(f(\lambda))\,\mathrm{d}\mu(\lambda). \tag{24}$$

*Equivalently, if $X$ is a random variable with distribution $\mu$, then $f(X)$ has distribution $f_*\mu$.*

**Proposition B.2.** *Assume that the dimensions of $\boldsymbol{M} \in \mathbb{R}^{d_x \times d_y}$ fulfill $d_x \leq d_y$ and let $r = d_x/d_y$. Let $\mu_{\boldsymbol{MM}^\top}$ be the expected spectral distribution of the random matrix $\boldsymbol{MM}^\top \in \mathbb{R}^{d_x \times d_x}$, and assume that it is absolutely continuous with respect to the Lebesgue measure. The expected spectral distribution of $\boldsymbol{A}$ is contained in the imaginary line and is given by*

$$\mu_{\boldsymbol{A}}(i\lambda) = \left(1 - \frac{2}{1 + \frac{1}{r}}\right)\delta_0(\lambda) + \frac{2|\lambda|}{1 + \frac{1}{r}}\mu_{\boldsymbol{MM}^\top}(\lambda^2). \tag{25}$$

*for $\lambda \in \mathbb{R}$. If $d_x \geq d_y$, then (25) holds with $\mu_{\boldsymbol{M}^\top\boldsymbol{M}}$ in place of $\mu_{\boldsymbol{MM}^\top}$ and $1/r$ in place of $r$.*

*Proof.* By the block determinant formula, we have that for $s \neq 0$,

$$\det\left(s\boldsymbol{I}_{d_1+d_2} - \boldsymbol{A}\right) = \begin{vmatrix} s\boldsymbol{I}_{d_1} & -\boldsymbol{M} \\ \boldsymbol{M}^\top & s\boldsymbol{I}_{d_2} \end{vmatrix} = \det(s\boldsymbol{I}_{d_2})\det(s\boldsymbol{I}_{d_1} + \boldsymbol{M}s^{-1}\boldsymbol{I}_{d_2}\boldsymbol{M}^\top)$$
$$= s^{d_2-d_1}\det(s^2\boldsymbol{I}_{d_1} + \boldsymbol{MM}^\top) \tag{26}$$

Thus, for every eigenvalue $-\lambda \leq 0$ of $-\boldsymbol{MM}^\top$, both $i\sqrt{\lambda}$ and $-i\sqrt{\lambda}$ are eigenvalues of $\boldsymbol{A}$. Since $\mathrm{rank}(\boldsymbol{MM}^\top) = \mathrm{rank}(\boldsymbol{M})$, we have $\mathrm{rank}(\boldsymbol{A}) = 2\mathrm{rank}(\boldsymbol{M})$. Thus, the rest of the eigenvalues of $\boldsymbol{A}$ are 0 and there is a total of $d - 2d_1 = d_2 - d_1$ of them. Notice that

$$\frac{d_1}{d_1 + d_2} = \frac{1}{\frac{d_1+d_2}{d_1}} = \frac{1}{1 + \frac{1}{r}} \tag{27}$$

Let $f_+(\lambda) = i\sqrt{\lambda}$, $f_-(\lambda) = -i\sqrt{\lambda}$, and let $(f_+)_*\mu_{\boldsymbol{MM}^\top}$ (resp., $(f_-)_*\mu_{\boldsymbol{MM}^\top}$) be the pushforward measure of $\mu_{\boldsymbol{MM}^\top}$ by the function $f_+$ (resp., $f_-$). Thus, by the definition of the pushforward measure (Definition 6),

$$\mu_{\boldsymbol{A}}(i\lambda) = \left(1 - \frac{2}{1 + \frac{1}{r}}\right)\delta_0(\lambda) + \frac{1}{1 + \frac{1}{r}}(f_+)_*\mu_{\boldsymbol{MM}^\top}(\lambda) + \frac{1}{1 + \frac{1}{r}}(f_-)_*\mu_{\boldsymbol{MM}^\top}(\lambda) \tag{28}$$

We compute the pushforwards $(f_+)_*\mu_{MM^\top}, (f_-)_*\mu_{MM^\top}$ performing the change of variables $y = \pm i\sqrt{\lambda}$ under the assumption that $\mu_{MM^\top}(\lambda) = \rho_{MM^\top}(\lambda)d\lambda$:

$$\int_{\mathbb{R}_{\geq 0}} g\left(\pm i\sqrt{\lambda}\right)\mathrm{d}\mu_{MM^\top}(\lambda) = \int_{\mathbb{R}_{\geq 0}} g\left(\pm i\sqrt{\lambda}\right)\rho_{MM^\top}(\lambda)d\lambda = \int_{\pm i\mathbb{R}_{\geq 0}} g(y)\,\rho_{MM^\top}(|y|^2)2|y|\,\mathrm{d}|y|, \tag{29}$$

which means that the density of $(f_+)_*\mu_{MM^\top}$ at $y \in i\mathbb{R}_{\geq 0}$ is $2|y|\rho_{MM^\top}(|y|^2)$ and the density of $(f_-)_*\mu_{MM^\top}$ at $y \in -i\mathbb{R}_{\geq 0}$ is also $2|y|\rho_{MM^\top}(|y|^2)$. $\square$

**Proposition B.3.** *The condition*

$$\forall P, Q \text{ polynomials } \langle P(\lambda), \lambda Q(\lambda)\rangle = 0 \implies \langle \lambda P(\lambda), Q(\lambda)\rangle = 0 \tag{30}$$

*is sufficient for any sequence $(P_k)_{k \geq 0}$ of orthogonal polynomials of increasing degrees to satisfy a three-term recurrence of the form*

$$\gamma_k P_k(\lambda) = (\lambda - \alpha_k)P_{k-1}(\lambda) - \beta_k P_{k-2}(\lambda), \tag{31}$$

*where*

$$\gamma_k = \frac{\langle \lambda P_{k-1}(\lambda), P_k(\lambda)\rangle}{\langle P_k(\lambda), P_k(\lambda)\rangle}, \quad \alpha_k = \frac{\langle \lambda P_{k-1}(\lambda), P_{k-1}(\lambda)\rangle}{\langle P_{k-1}(\lambda), P_{k-1}(\lambda)\rangle}, \quad \beta_k = \frac{\langle \lambda P_{k-1}(\lambda), P_{k-2}(\lambda)\rangle}{\langle P_{k-2}(\lambda), P_{k-2}(\lambda)\rangle} \tag{32}$$

*Proof.* Since $\lambda P_{k-1}(\lambda)$ is a polynomial of degree $k$, and $(P_j)_{0 \le j \le k}$ is a basis of the polynomials of degree up to $k$, we can write

$$\lambda P_{k-1}(\lambda) = \sum_{j=0}^{k} \frac{\langle \lambda P_{k-1}, P_j \rangle}{\langle P_j, P_j \rangle} P_j(\lambda) \tag{33}$$

Now, remark that for all $j < k - 2$, $\langle P_{k-1}, \lambda P_j \rangle = 0$ because the inner product of $P_{k-1}$ with a polynomial of degree at most $k - 2$. If we make use of the condition (30), this implies that $\langle \lambda P_{k-1}, P_j \rangle = 0$ for all $j < k - 2$. Plugging this into (33), we obtain (31). $\qquad\square$

**Proposition B.4.** *Let $\Pi_t^{\mathbb{R}}$ be the set of polynomials with real coefficients and degree at most $t$. For $t \ge 0$ even, the minimum of the problem*

$$\min_{P_t \in \Pi_t^{\mathbb{R}}, P_t(0)=1} \int_{i\mathbb{R} \setminus \{0\}} |P_t(\lambda)|^2 |\lambda| \rho_{\boldsymbol{MM}^\top}(|\lambda|^2) \, \mathrm{d}|\lambda| \tag{34}$$

*is attained by an even polynomial with real coefficients.*

*Proof.* Since $\mathrm{d}\mu(i\lambda) \overset{\text{def}}{=} |\lambda| \rho_{MM^\top}(|\lambda|^2) \, \mathrm{d}|\lambda|$ is supported in the imaginary axis and is symmetric with respect to 0, for all polynomials $P, Q$,

$$\langle \lambda P(\lambda), Q(\lambda) \rangle = \int_{i\mathbb{R}} \lambda P(\lambda) Q(\lambda)^* d\mu(\lambda) = -\int_{i\mathbb{R}} P(\lambda) \lambda^* Q(\lambda)^* d\mu(\lambda) = -\langle P(\lambda), \lambda Q(\lambda) \rangle. \tag{35}$$

Hence, $\langle P(\lambda), \lambda Q(\lambda) \rangle = 0$ implies $\langle \lambda P(\lambda), Q(\lambda) \rangle = 0$. By Proposition B.3, a three-term recurrence (31) and (32) for the orthonormal sequence $(\phi_t)_{t \ge 0}$ of polynomials holds.

By Proposition B.5, the orthonormal polynomials $(\phi_t)_{t \ge 0}$ of even (resp. odd) degree are even (resp. odd) and have real coefficients. Hence, for all $t \ge 0$ even

$$\frac{\sum_{k=0}^{t} \phi_k(\lambda) \phi_k(0)^*}{\sum_{k=0}^{t} |\phi_k(0)|^2} = \frac{\sum_{k=0}^{t/2} \phi_{2k}(\lambda) \phi_{2k}(0)^*}{\sum_{k=0}^{t/2} |\phi_{2k}(0)|^2} \tag{36}$$

is an even polynomial with real coefficients. By Theorem 2.3, this polynomial attains the minimum of the problem

$$\min_{P_t \in \Pi_t^{\mathbb{C}}, P_t(0)=1} \int_{i\mathbb{R} \setminus \{0\}} |P_t(\lambda)|^2 |\lambda| \rho_{MM^\top}(|\lambda|^2) \, \mathrm{d}|\lambda| \tag{37}$$

and, a fortiori, the minimum of the problem in (34), in which the minimization is restricted polynomials with real coefficients instead of complex coefficients. $\qquad\square$

**Proposition B.5.** *The polynomials $(\phi_t)_{t \ge 0}$ of the orthonormal sequence corresponding to the measure $\mu(i\lambda) = |\lambda| \rho_{MM^\top}(|\lambda|^2) d|\lambda|$ have real coefficients and are even (resp. odd) for even (resp. odd) $k$.*

*Proof.* The proof is by induction. The base case follows from the choice $\phi_0 = 1$. Assuming that $\phi_{k-1} \in \mathbb{R}[X]$ by the induction hypothesis, we show that $\alpha_k = 0$ (where $\alpha_k$ is the coefficient from (31) and (32)):

$$\langle \lambda \phi_{k-1}(\lambda), \phi_{k-1}(\lambda) \rangle = \int_{i\mathbb{R}} \lambda |\phi_{k-1}(\lambda)|^2 |\lambda| \rho_{MM^\top}(|\lambda|^2) d|\lambda|$$
$$= \int_{\mathbb{R}_{\ge 0}} i\lambda(|\phi_{k-1}(i\lambda)|^2 - |\phi_{k-1}(-i\lambda)|^2) \lambda \rho_{MM^\top}(\lambda^2) d\lambda = 0 \tag{38}$$

The last equality follows from $|\phi_{k-1}(i\lambda)|^2 = |\phi_{k-1}(-i\lambda)|^2$, which holds because $\phi_{k-1}(i\lambda)^* = \phi_{k-1}(-i\lambda)$, and in turn this is true because $\phi_{k-1} \in \mathbb{R}[X]$ by the induction hypothesis.

Once we have seen that $\alpha_k = 0$, it is straightforward to apply the induction hypothesis once again to show that $\phi_k$ also satisfies the even/odd property. Namely, for $k$ even (resp. odd), $\gamma_k P_k = \lambda P_{k-1} - \beta_k P_{k-2}$, and the two polynomials in the right-hand side have even (resp. odd) degrees.

Finally, $\phi_k$ must have real coefficients because $\phi_{k-1}$ and $\phi_{k-2}$ have real coefficients by the induction hypothesis, and the recurrence coefficient $\beta_k$ is real, as

$$
\begin{aligned}
\langle \lambda P_{k-1}(\lambda), P_{k-2}(\lambda) \rangle &= \int_{i\mathbb{R}} \lambda \phi_{k-1}(\lambda) \phi_{k-2}(\lambda)^* |\lambda| \rho_{MM^\top}(|\lambda|^2) d|\lambda| \\
&= \int_{\mathbb{R}_{\geq 0}} i\lambda (\phi_{k-1}(i\lambda) \phi_{k-2}(i\lambda)^* - \phi_{k-1}(i\lambda)^* \phi_{k-2}(i\lambda)) \lambda \rho_{MM^\top}(\lambda^2) d\lambda \\
&= - \int_{\mathbb{R}_{\geq 0}} 2\lambda \mathrm{Im}(\phi_{k-1}(i\lambda)\phi_{k-2}(i\lambda)^*) \lambda \rho_{MM^\top}(\lambda^2) d\lambda \in \mathbb{R}.
\end{aligned}
\tag{39}
$$

$\square$

**Proposition B.6.** *Let $t \geq 0$ even. Assume that on $\mathbb{R}_{>0}$, the expected spectral density $\mu_{MM^\top}$ has Radon-Nikodym derivative $\rho_{MM^\top}$ with respect to the Lebesgue measure. If*

$$
Q_{t/2}^\star \stackrel{def}{=} \underset{\substack{P_{t/2} \in \Pi_{t/2}^{\mathbb{R}}, \\ P_{t/2}(0)=1}}{\arg\min} \int_{\mathbb{R}_{>0}} P_{t/2}(\lambda)^2 \, d\mu_{-A^2}(\lambda),
\tag{40}
$$

*and*

$$
P_t^\star \stackrel{def}{=} \underset{\substack{P_t \in \Pi_t^{\mathbb{R}}, \\ P_t(0)=1}}{\arg\min} \int_{i\mathbb{R} \backslash \{0\}} |P_t(\lambda)|^2 |\lambda| \rho_{MM^\top}(|\lambda|^2) \, d|\lambda|,
\tag{41}
$$

*then $P_t^\star(\lambda) = Q_{t/2}^\star(-\lambda^2)$.*

*Proof.* First, remark that the equalities in (40) and (41) are well defined because the $\arg\min$ are unique by Theorem 2.3. Without loss of generality, assume that $d_x \leq d_y$ (otherwise switch the players), and let $r \stackrel{def}{=} d_x/d_y < 1$. Since,

$$
-A^2 = \begin{bmatrix} MM^\top & 0 \\ 0 & M^\top M \end{bmatrix},
\tag{42}
$$

each eigenvalue of $MM^\top \in \mathbb{R}^{d_x \times d_x}$ is an eigenvalue of $-A^2$ with doubled duplicity, and the rest of eigenvalues are zero. Hence, we have $\mu_{-A^2} = \left(1 - 2/(1 + \frac{1}{r})\right)\delta_0 + 2\mu_{MM^\top}/(1 + \frac{1}{r})$. Thus, for all $t \geq 0$,

$$
Q_t^\star = \underset{\substack{P_t \in \Pi_t^{\mathbb{R}}, \\ P_t(0)=1}}{\arg\min} \int_{\mathbb{R}_{>0}} P_t(\lambda)^2 \, d\mu_{-A^2}(\lambda) = \underset{\substack{P_t \in \Pi_t^{\mathbb{R}}, \\ P_t(0)=1}}{\arg\min} \int_{\mathbb{R}_{>0}} P_t(\lambda)^2 \rho_{MM^\top}(\lambda) \, d\lambda
\tag{43}
$$

By Proposition B.4, for an even $t \geq 0$ the minimum in (41) is attained by an even polynomial with real coefficients. Hence,

$$
\begin{aligned}
\underset{\substack{P_t \in \Pi_t^{\mathbb{R}}, \\ P_t(0)=1}}{\min} \int_{i\mathbb{R} \backslash \{0\}} |P_t(\lambda)|^2 |\lambda| \rho_{MM^\top}(|\lambda|^2) \, d|\lambda| &= \underset{\substack{P_{t/2} \in \Pi_{t/2}^{\mathbb{R}}, \\ P_{t/2}(0)=1}}{\min} \int_{i\mathbb{R} \backslash \{0\}} |P_{t/2}(\lambda^2)|^2 |\lambda| \rho_{MM^\top}(|\lambda|^2) \, d|\lambda| \\
&= 2 \underset{\substack{P_{t/2} \in \Pi_{t/2}^{\mathbb{R}}, \\ P_{t/2}(0)=1}}{\min} \int_{\mathbb{R}_{>0}} |P_{t/2}((i\lambda)^2)|^2 \lambda \rho_{MM^\top}(\lambda^2) \, d\lambda = 2 \underset{\substack{P_{t/2} \in \Pi_{t/2}^{\mathbb{R}}, \\ P_{t/2}(0)=1}}{\min} \int_{\mathbb{R}_{>0}} P_{t/2}(\lambda^2)^2 \lambda \rho_{MM^\top}(\lambda^2) \, d\lambda \\
&= \underset{\substack{P_{t/2} \in \Pi_{t/2}^{\mathbb{R}}, \\ P_{t/2}(0)=1}}{\min} \int_{\mathbb{R}_{>0}} P_{t/2}(\lambda)^2 \rho_{MM^\top}(\lambda) \, d\lambda
\end{aligned}
\tag{44}
$$

Moreover, for any polynomial $Q_{t/2}$ that attains the minimum on the right-most term, the polynomial $P_t(\lambda) = Q_{t/2}(-\lambda^2)$ attains the minimum on the left-most term. In particular, using (43), $P_t^\star(\lambda) \stackrel{def}{=} Q_{t/2}^\star(-\lambda^2)$ attains the minimum on the left-most term. $\square$

**Theorem 3.1.** *Suppose that Assumption 1 holds and that the expected spectral distribution of* $MM^\top$ *is absolutely continuous with respect to the Lebesgue measure. Then, the method* (11) *is average-case optimal for bilinear games when* $h_t$, $m_t$ *are chosen to be the coefficients of the average-case optimal minimization of* $\frac{1}{2}\|F(x)\|^2$.

*Proof.* Making use of Theorem 2.1 and Proposition B.2, we obtain that for any first-order method using the vector field $F$,

$$\mathbb{E}[\text{dist}(x_t, \mathcal{X}^\star)] = R^2 \int_{\mathbb{C}\setminus\{0\}} |P_t(\lambda)|^2 \, d\mu_A(\lambda) = \frac{2R^2}{1 + \frac{1}{r}} \int_{i\mathbb{R}\setminus\{0\}} |P_t(\lambda)|^2 |\lambda| \rho_{MM^\top}(|\lambda|^2) \, d|\lambda| \tag{45}$$

Let $Q^\star_{t/2}, P^\star_t$ be as defined in (41) and (40). For $t \geq 0$ even the iteration $t$ of the average-case optimal method for the bilinear game must satisfy

$$x_t - P_{\mathcal{X}^\star}(x_0) = P^\star_t(A)(x_0 - P_{\mathcal{X}^\star}(x_0)) = Q^\star_{t/2}(-A^2)(x_0 - P_{\mathcal{X}^\star}(x_0)) \tag{46}$$

On the other hand, the first-order methods for the minimization of the function $\frac{1}{2}\|F(x)\|^2$ make use of the vector field $\nabla\left(\frac{1}{2}\|F(x)\|^2\right) = A^\top(Ax + b) = -A^2(x - x^\star)$. Let $\mu_{-A^2}$ be the spectral density of $-A^2$. By Theorem 2.1, the average-case optimal first-order method for the minimization problem is the one for which the residual polynomial $P_t$ (Proposition 2.1) minimizes the functional $\int_{\mathbb{R}} P_t^2 \, d\mu_{-A^2}$. That is, the residual polynomial is $Q^\star_t$. From (46), we see that the $t$-th iterate of the average-case optimal method for $F$ is equal to the $t/2$-th iterator of the average-case optimal method for $\nabla\left(\frac{1}{2}\|F(x)\|^2\right)$. □

## C PROOFS OF THEOREM 4.1 AND THEOREM 4.2

**Theorem 4.1.** *Under Assumption 4 and the assumptions of Theorem 2.1, the following algorithm is optimal in the average case, with* $y_{-1} = y_0 = x_0$:

$$y_t = a_t y_{t-1} + (1 - a_t) y_{t-2} + b_t F(y_{t-1})$$
$$x_t = \frac{B_t}{B_t + \beta_t} x_{t-1} + \frac{\beta_t}{B_t + \beta_t} y_t, \quad \beta_t = \phi_t^2(0), \quad B_t = B_{t-1} + \beta_{t-1}, \quad B_0 = 0. \tag{16}$$

*where* $(\phi_k(0))_{k\geq 0}$ *can be computed using the three-term recurrence (upon normalization). Moreover,* $\mathbb{E}_{(A, x^\star, x_0)} \text{dist}(x_t, \mathcal{X}^\star)$ *converges to zero at rate* $1/B_t$.

*Proof.* We prove by induction that

$$x_t - x^\star = \frac{\sum_{k=0}^t \phi_k(A)\phi_k(0)^*}{\sum_{k=0}^t \phi_k(0)^2}(x_0 - x^\star) \tag{47}$$

The base step $t = 0$ holds trivially because $\phi_0 = 1$. Assume that (47) holds for $t - 1$. Subtracting $x^\star$ from (16), we have

$$x_t - x^\star = \frac{\sum_{k=0}^{t-1} \phi_k(0)^2}{\sum_{k=0}^t \phi_k(0)^2}(x_{t-1} - x^\star) + \frac{\phi_t(0)^2}{\sum_{k=0}^t \phi_k(0)^2}(y_t - x^\star) \tag{48}$$

If

$$\phi_t(0)^2(y_t - x^\star) = \phi_t(0)\phi_t(A)(x_0 - x^\star), \tag{49}$$

by the induction hypothesis for $t - 1$ and (48), we have

$$x_t - x^\star = \frac{\sum_{k=0}^{t-1} \phi_t(0)\phi_t(A)}{\sum_{k=0}^t \phi_k(0)^2}(x_0 - x^\star) + \frac{\phi_t(0)\phi_t(A)}{\sum_{k=0}^t \phi_k(0)^2}(x_0 - x_*)$$
$$= \frac{\sum_{k=0}^t \phi_t(0)\phi_t(A)}{\sum_{k=0}^t \phi_k(0)^2}(x_0 - x_*), \tag{50}$$

which concludes the proof of (47). The only thing left is to show (49), again by induction. The base case follows readily from $\boldsymbol{y}_0 = \boldsymbol{x}_0$ in (16). Dividing by $\phi_t(0)^2$, we rewrite (49) as

$$\boldsymbol{y}_t - \boldsymbol{x}^\star = \frac{\phi_t(\boldsymbol{A})}{\phi_t(0)}(\boldsymbol{x}_0 - \boldsymbol{x}^\star) = \psi_t(\boldsymbol{A})(\boldsymbol{x}_0 - \boldsymbol{x}^\star), \tag{51}$$

where $\psi_t$ is the $t$-th orthogonal residual polynomial of sequence. By Assumption 4, $\psi_t$ must satisfy the recurrence in (15). If we subtract $x_*$ from the second line of (16), we apply the induction hypothesis and then the recurrence in (15), we obtain

$$
\begin{aligned}
\boldsymbol{y}_t - \boldsymbol{x}^\star &= a_t(\boldsymbol{y}_{t-1} - \boldsymbol{x}^\star) + (1 - a_t)(\boldsymbol{y}_{t-2} - \boldsymbol{x}^\star) + b_t F(\boldsymbol{y}_{t-1}) \\
&= a_t(\boldsymbol{y}_{t-1} - \boldsymbol{x}^\star) + (1 - a_t)(\boldsymbol{y}_{t-2} - \boldsymbol{x}^\star) + b_t \boldsymbol{A}(\boldsymbol{y}_{t-1} - \boldsymbol{x}_*) \\
&= a_t \psi_{t-1}(\boldsymbol{A})(\boldsymbol{x}_0 - \boldsymbol{x}^\star) + (1 - a_t)\psi_{t-2}(\boldsymbol{A})(\boldsymbol{x}_0 - \boldsymbol{x}^\star) + b_t \boldsymbol{A}\psi_{t-1}(\boldsymbol{A})(\boldsymbol{x}_0 - \boldsymbol{x}^\star) \\
&= \psi_t(\boldsymbol{A})(\boldsymbol{x}_0 - \boldsymbol{x}^\star),
\end{aligned}
\tag{52}
$$

thus concluding the proof of (49). $\qquad\square$

**Proposition C.1.** *Suppose that Assumption 5 holds with $C = 0$, that is, the circular support of $\mu$ is centered at $0$. Then, the basis of orthonormal polynomials for the scalar product*

$$\langle P, Q \rangle = \int_{D_{R,0}} P(\lambda)Q(\lambda)^* \, \mathrm{d}\mu(\lambda) \quad is \quad \phi_k(\lambda) = \frac{\lambda^k}{D_{k,R}}, \quad \forall k \geq 0, \tag{53}$$

*where* $K_{k,R} = \sqrt{2\pi \int_0^R r^{2k} d\mu_R(r)}.$

*Proof.* First, we will show that if $\mu$ satisfies Assumption 5 with $C = 0$, then $\langle \lambda^i, \lambda^j \rangle = 0$ if $j, k \geq 0$ with $j \neq k$ (without loss of generality, suppose that $j > k$).

$$
\begin{aligned}
\langle \lambda^j, \lambda^k \rangle &= \int_{D_{R,0}} \lambda^j (\lambda^*)^k \, \mathrm{d}\mu(\lambda) = \int_{D_{R,0}} \lambda^{j-k} |\lambda|^{2k} \, \mathrm{d}\mu(\lambda) \\
&= \int_0^R \frac{1}{2\pi} \int_0^{2\pi} (re^{i\theta})^{j-k} r^{2k} \, \mathrm{d}\theta \, \mathrm{d}\mu_R(r) = \frac{1}{2\pi} \int_0^{2\pi} e^{i\theta(j-k)} \, \mathrm{d}\theta \int_0^R r^{j+k} \, \mathrm{d}\mu_R(r) \\
&= \frac{e^{i2\pi} - 1}{2\pi i(j-k)} \int_0^R r^{j+k} \, \mathrm{d}\mu_R(r) = 0
\end{aligned}
\tag{54}
$$

And for all $k \geq 0$,

$$\langle \lambda^k, \lambda^k \rangle = \int_{D_{R,0}} |\lambda^k|^2 \, \mathrm{d}\mu(\lambda) = \int_0^R \frac{1}{2\pi} \int_0^{2\pi} r^{2k} \, \mathrm{d}\theta \, \mathrm{d}\mu_R(r) = \int_0^{2\pi} r^{2k} \, \mathrm{d}\mu_R(r). \tag{55}$$

$\qquad\square$

**Proposition 4.1.** *If $\mu$ satisfies Assumption 5, the sequence of orthonormal polynomials is $(\phi_t)_{t \geq 0}$,*

$$\phi_t(\lambda) = \frac{(\lambda - C)^t}{K_{t,R}}, \quad where \quad K_{t,R} = \sqrt{\int_0^R r^{2t} \, \mathrm{d}\mu_R(r)}. \tag{17}$$

*Proof.* The result follows from Proposition C.1 using the change of variables $z \to z + C$. To compute the measure $\mu_R$ for the uniform measure on $D_{C,R}$, we perform a change of variables to circular coordinates:

$$
\begin{aligned}
\int_{D_{C,R}} f(\lambda) \, \mathrm{d}\mu(\lambda) &= \frac{1}{\pi R^2} \int_0^R \int_0^{2\pi} f(C + re^{i\theta})r \, \mathrm{d}\theta \, \mathrm{d}r = \int_0^R \int_0^{2\pi} f(C + re^{i\theta}) \, \mathrm{d}\theta \, \mathrm{d}\mu_R(r). \\
&\implies \mathrm{d}\mu_R(r) = \frac{r}{\pi R^2} \, \mathrm{d}r
\end{aligned}
\tag{56}
$$

And

$$\int_0^R r^{2t} \, \mathrm{d}\mu_R(r) = \frac{1}{\pi R^2} \int_0^R r^{2t+1} \, \mathrm{d}r = \frac{1}{\pi} \frac{R^{2t}}{2t+2} \implies K_{t,R} = R^t/\sqrt{t+1}. \tag{57}$$

$\qquad\square$

**Theorem 4.2.** *Given an initialization $x_0 (y_0 = x_0)$, if Assumption 5 is fulfilled with $R < C$ and the assumptions of Theorem 2.1 hold, then the average-case optimal first-order method is*

$$y_t = y_{t-1} - \frac{1}{C} F(y_{t-1}), \quad \beta_t = C^{2t}/K_{t,R}^2, \quad B_t = B_{t-1} + \beta_{t-1},$$
$$x_t = \frac{B_t}{B_t + \beta_t} x_{t-1} + \frac{\beta_t}{B_t + \beta_t} y_t. \tag{18}$$

*Moreover, $\mathbb{E}_{(A, x^\star, x_0)} \operatorname{dist}(x_t, \mathcal{X}^\star)$ converges to zero at rate $1/B_t$.*

*Proof.* By Proposition 4.1, the sequence of residual orthogonal polynomials is given by $\psi_t(\lambda) = \phi_t(\lambda)/\phi_t(0) = \left(1 - \frac{\lambda}{C}\right)^t$. Hence, Assumption 4 is fulfilled with $a_t = 1, b_t = -\frac{1}{C}$, as $\psi_t(\lambda) = \psi_{t-1}(\lambda) - \frac{\lambda}{C}\psi_{t-1}(\lambda)$. We apply Theorem 4.1 and make use of the fact that $\phi_k(0)^2 = \frac{C^{2k}}{K_{t,R}^2}$. See Proposition D.3 for the rate on $\operatorname{dist}(x_t, \mathcal{X}^\star)$. $\qquad \square$

# D    PROOF OF PROPOSITION 5.2

**Proposition D.1.** *Suppose that the assumptions of Theorem 4.2 hold with the probability measure $\mu_R$ fulfilling $\mu_R([r, R]) = \Omega((R - r)^\kappa)$ for $r$ in $[r_0, R]$ for some $r_0 \in [0, R)$ and for some $\kappa \in \mathbb{Z}$. Then,*

$$\lim_{t \to \infty} \frac{\frac{C^{2t}}{K_{t,R}^2}}{\sum_{k=0}^{t} \frac{C^{2k}}{K_{k,R}^2}} = 1 - \frac{R^2}{C^2}. \tag{58}$$

*Proof.* Given $\epsilon > 0$, let $c_\epsilon \in \mathbb{Z}_{\geq 0}$ be the minimum such that

$$\frac{1}{\sum_{i=0}^{c_\epsilon} \left(\frac{R^2}{C^2}\right)^i} \leq (1 + \epsilon) \frac{1}{\sum_{i=0}^{\infty} \left(\frac{R^2}{C^2}\right)^i} = (1 + \epsilon) \left(1 - \frac{R^2}{C^2}\right) \tag{59}$$

Define $Q_{t,R} \stackrel{\text{def}}{=} \frac{R^{2t}}{K_{t,R}^2}$. Then,

$$\frac{\frac{C^{2t}}{K_{t,R}^2}}{\sum_{k=0}^{t} \frac{C^{2k}}{K_{k,R}^2}} = \frac{\frac{C^{2t}}{R^{2t}} Q_{t,R}}{\sum_{k=0}^{t} \frac{C^{2k}}{R^{2k}} Q_{k,R}} = \frac{Q_{t,R}}{\sum_{k=0}^{t} \left(\frac{R^2}{C^2}\right)^{t-k} Q_{k,R}} \tag{60}$$

Now, on one hand, using that $Q_{t,R}$ is an increasing sequence on $t$,

$$\frac{Q_{t,R}}{\sum_{k=0}^{t} \left(\frac{R^2}{C^2}\right)^{t-k} Q_{k,R}} \geq \frac{1}{\sum_{k=0}^{t} \left(\frac{R^2}{C^2}\right)^{t-k}} \geq \frac{1}{\sum_{k=0}^{\infty} \left(\frac{R^2}{C^2}\right)^k} = 1 - \frac{R^2}{C^2} \tag{61}$$

On the other hand, for $t \geq c_\epsilon$,

$$\frac{Q_{t,R}}{\sum_{k=0}^{t} \left(\frac{R^2}{C^2}\right)^{t-k} Q_{k,R}} \leq \frac{Q_{t,R}}{\sum_{k=t-c_\epsilon}^{t} \left(\frac{R^2}{C^2}\right)^{t-k} Q_{k,R}} = \frac{Q_{t,R}}{\sum_{k=t-c_\epsilon}^{t} \left(\frac{R^2}{C^2}\right)^{t-k} \left(Q_{t,R} - \int_k^t \frac{d}{ds} Q_{s,R} \, ds\right)} \tag{62}$$

Thus, we want to upper-bound $\int_k^t \frac{d}{ds} Q_{s,R} \, ds$. First, notice that

$$\frac{d}{ds} Q_{s,R} = \frac{d}{ds} \left(\int_0^R \left(\frac{r}{R}\right)^{2s} d\mu_R(r)\right)^{-1} = \frac{\int_0^R \left(\frac{r}{R}\right)^{2s} \left(-\log(\frac{r}{R})\right) d\mu_R(r)}{\left(\int_0^R \left(\frac{r}{R}\right)^{2s} d\mu_R(r)\right)^2} \tag{63}$$

By concavity of the logarithm function we obtain $\log(\frac{R}{r}) \leq \frac{R}{r_0} - 1$ for $r \in [r_0, R]$. Choose $r_0$ close enough to $R$ so that $\frac{R}{r_0} - 1 \leq \epsilon/c_\epsilon$. We obtain that

$$\int_0^R \left(\frac{r}{R}\right)^{2s} \log\left(\frac{R}{r}\right) d\mu_R(r) \leq \int_0^{r_0} \left(\frac{r}{R}\right)^{2s} \log\left(\frac{R}{r}\right) d\mu_R(r) + \int_{r_0}^R \left(\frac{r}{R}\right)^{2s} \left(\frac{R}{r_0} - 1\right) d\mu_R(r). \tag{64}$$

Thus,

$$\int_k^t \frac{d}{ds} Q_{s,R} \, ds \le \int_k^t \frac{\int_0^{r_0} \left(\frac{r}{R}\right)^{2s} \log\left(\frac{R}{r}\right) \, d\mu_R(r)}{\left(\int_0^R \left(\frac{r}{R}\right)^{2s} d\mu_R(r)\right)^2} \, ds + \int_k^t \frac{\int_{r_0}^R \left(\frac{r}{R}\right)^{2s} \left(\frac{R}{r_0} - 1\right) d\mu_R(r)}{\left(\int_0^R \left(\frac{r}{R}\right)^{2s} d\mu_R(r)\right)^2} \, ds. \quad (65)$$

Using that $\log x \le x$, for $k \in [t - c_\epsilon, t]$ we can bound the first term of (65) as

$$\begin{aligned}
\int_k^t \frac{\int_0^{r_0} \left(\frac{r}{R}\right)^{2s} \log\left(\frac{R}{r}\right) d\mu_R(r)}{\left(\int_0^R \left(\frac{r}{R}\right)^{2s} d\mu_R(r)\right)^2} \, ds &\le \int_k^t \frac{\int_0^{r_0} \left(\frac{r}{R}\right)^{2s-1} d\mu_R(r)}{\left(\int_0^R \left(\frac{r}{R}\right)^{2s} d\mu_R(r)\right)^2} \, ds \\
&\le (t-k) \frac{\left(\frac{r_0}{R}\right)^{2k-1}}{\left(\int_0^R \left(\frac{r}{R}\right)^{2t} d\mu_R(r)\right)^2} \\
&\le c_\epsilon \left(\frac{r_0}{R}\right)^{2(t-c_\epsilon)-1} Q_{t,R}^2 \\
&\le c_\epsilon \left(\frac{r_0}{R}\right)^{2(t-c_\epsilon)-1} \frac{1}{(c_1)^2} (2t+1)^{2\kappa} \xrightarrow{t\to\infty} 0.
\end{aligned} \quad (66)$$

In the last inequality we use that by Proposition D.2, for $t$ large enough, $Q_{t,R} = \frac{R^{2t}}{K_{t,R}^2} \le (2t + 1)^k / c_1$. For $k \in [t - c_\epsilon, t]$, the second term of (65) can be bounded as

$$\begin{aligned}
\int_k^t \frac{\int_{r_0}^R \left(\frac{r}{R}\right)^{2s} \frac{R}{r_0} \, d\mu_R(r)}{\left(\int_0^R \left(\frac{r}{R}\right)^{2s} d\mu_R(r)\right)^2} \, ds &\le (t-k) \left(\frac{R}{r_0} - 1\right) \frac{1}{\int_0^R \left(\frac{r}{R}\right)^{2t} d\mu_R(r)} \\
&\le c_\epsilon \left(\frac{R}{r_0} - 1\right) \frac{1}{\int_0^R \left(\frac{r}{R}\right)^{2t} d\mu_R(r)} \\
&\le \epsilon Q_{t,R}.
\end{aligned} \quad (67)$$

From (65), (66) and (67), we obtain that for $t$ large enough, for $k \in [t - c_\epsilon, t]$,

$$\int_k^t \frac{d}{ds} Q_{s,R} \, ds \le 2\epsilon Q_{t,R}. \quad (68)$$

Hence, we can bound the right-hand side of (62):

$$\begin{aligned}
\frac{Q_{t,R}}{\sum_{k=t-c_\epsilon}^t \left(\frac{R^2}{C^2}\right)^{t-k} \left(Q_{t,R} - \int_k^t \frac{d}{ds} Q_{s,R} \, ds\right)} &\le \frac{Q_{t,R}}{\sum_{k=t-c_\epsilon}^t \left(\frac{R^2}{C^2}\right)^{t-k} (Q_{t,R} - 2\epsilon Q_{t,R})} \\
= \frac{1}{(1-2\epsilon) \sum_{k=t-c_\epsilon}^t \left(\frac{R^2}{C^2}\right)^{t-k}} &= \frac{1}{(1-2\epsilon) \sum_{k=0}^{c_\epsilon} \left(\frac{R^2}{C^2}\right)^k} \le \frac{1+\epsilon}{1-2\epsilon} \left(1 - \frac{R^2}{C^2}\right).
\end{aligned} \quad (69)$$

The last inequality follows from the definition of $c_\epsilon$ in (59). Since $\epsilon$ is arbitrary, by the sandwich theorem applied on (60), (61) and (69),

$$\lim_{t\to\infty} \frac{\frac{C^{2t}}{K_{t,R}^2}}{\sum_{k=0}^t \frac{C^{2k}}{K_{k,R}^2}} = 1 - \frac{R^2}{C^2}. \quad (70)$$

$\square$

**Proposition D.2.** *Under the assumptions of Theorem 4.2, we have that there exists $c_1 > 0$ such that for $t$ large enough,*

$$K_{t,R}^2 \ge c_1 R^{2t} (2t+1)^{-\kappa}. \quad (71)$$

*Proof.* By the assumption on $\mu_R$, there exist $r_0, c_1, \kappa > 0$ such that

$$
\begin{aligned}
K_{t,R}^2 &\overset{\text{def}}{=} 2\pi \int_0^R r^{2t}\,\mathrm{d}\mu_R(r) = 2\pi \int_0^{r_0} r^{2t}\,\mathrm{d}\mu_R(r) + 2\pi \int_{r_0}^R r^{2t}\,\mathrm{d}\mu_R(r) \\
&\geq 2\pi c_1 \int_{r_0}^R r^{2t}(R-r)^{\kappa-1}\,\mathrm{d}r = -2\pi c_1 \int_0^{r_0} r^{2t}(R-r)^{\kappa-1}\,\mathrm{d}r + 2\pi c_1 \int_0^R r^{2t}(R-r)^{\kappa-1}\,\mathrm{d}r \\
&\geq -2\pi c_1 R r_0^{2t} + 2\pi c_1 R^{2t+\kappa} B(2t+1, \kappa).
\end{aligned}
\tag{72}
$$

where the beta function $B(x,y)$ is defined as

$$
B(x,y) \overset{\text{def}}{=} \int_0^1 r^{x+1}(1-r)^{y+1}\,\mathrm{d}r.
\tag{73}
$$

Using the link between the beta function and the gamma function $B(x,y) = \Gamma(x)\Gamma(y)/\Gamma(x+y)$, and Stirling's approximation, we obtain that for fixed $y$ and large $x$,

$$
B(x,y) \sim \Gamma(y)x^{-y}.
\tag{74}
$$

Hence, for $t$ large enough, $B(2t+1,\kappa) \sim \Gamma(\kappa)(2t+1)^{-\kappa} = (\kappa-1)!(2t+1)^{-\kappa}$. Hence, from (72) we obtain that there exist $c_1'$ depending only on $\kappa$ and $r_0$ such that for $t$ large enough

$$
K_{t,R}^2 \geq -2\pi c_1 R r_0^{2t} + 2\pi c_1 R^{2t+\kappa}(k-1)!(2t+1)^{-\kappa} \geq c_1' R^{2t}(2t+1)^{-\kappa}.
\tag{75}
$$

$\square$

**Proposition 5.2.** *Suppose that the assumptions of Theorem 4.2 hold with $\mu_R \in \mathcal{P}([0,R])$ fulfilling $\mu_R([r,R]) = \Omega((R-r)^\kappa)$ for $r$ in $[r_0, R]$ for some $r_0 \in [0,R)$ and for some $\kappa \in \mathbb{Z}$. Then, the average-case asymptotically optimal algorithm is, with $\boldsymbol{y}_0 = \boldsymbol{x}_0$:*

$$
\begin{aligned}
\boldsymbol{y}_t &= \boldsymbol{y}_{t-1} - \tfrac{1}{C}F(\boldsymbol{y}_{t-1}), \\
\boldsymbol{x}_t &= \left(\tfrac{R}{C}\right)^2 \boldsymbol{x}_{t-1} + \left(1 - \left(\tfrac{R}{C}\right)^2\right)\boldsymbol{y}_t.
\end{aligned}
\tag{20}
$$

*Moreover, the convergence rate for this algorithm is asymptotically the same one as for the optimal algorithm in Theorem 4.2. Namely, $\lim_{t\to\infty} \mathbb{E}\left[\mathrm{dist}(\boldsymbol{x}_t, \mathcal{X}^\star)\right]B_t = 1$.*

*Proof.* The proof follows directly from Theorem 4.2 and Proposition D.1. See (77) and (79) in Proposition D.3 for the statement regarding the convergence rate. $\square$

**Proposition D.3.** *For the average-case optimal algorithm* (18),

$$
\mathbb{E}\,\mathrm{dist}(\boldsymbol{x}_t, \mathcal{X}^\star) = \xi_{opt}(t) \overset{\text{def}}{=} \frac{1}{\sum_{k=0}^t \frac{C^{2k}}{K_{k,R}^2}}
\tag{76}
$$

*For the average-case asymptotically optimal algorithm* (20),

$$
\mathbb{E}\,\mathrm{dist}(\boldsymbol{x}_t, \mathcal{X}^\star) = \xi_{asymp}(t) \overset{\text{def}}{=} \left(1 - \left(\frac{R}{C}\right)^2\right)^2 \sum_{k=1}^t \frac{K_{k,R}^2}{C^{2k}}\left(\frac{R}{C}\right)^{4(t-k)} + \left(\frac{R}{C}\right)^{4t}
\tag{77}
$$

*For the iterates $\boldsymbol{y}_t$ in* (18), *i.e. gradient descent with stepsize $1/C$, we have*

$$
\mathbb{E}\,\mathrm{dist}(\boldsymbol{y}_t, \mathcal{X}^\star) = \xi_{GD}(t) \overset{\text{def}}{=} \frac{K_{t,R}^2}{C^{2t}}
\tag{78}
$$

*Moreover, for all $t \geq 0$, we have $\xi_{opt}(t) \leq \xi_{asymp}(t)$, and under the assumptions of* (5.1),

$$
\lim_{t\to\infty} \frac{\xi_{opt}(t)}{\xi_{asymp}(t)} = 1, \quad \lim_{t\to\infty} \frac{\xi_{opt}(t)}{\xi_{GD}(t)} = \frac{\xi_{asymp}(t)}{\xi_{GD}(t)} = 1 - \left(\frac{R}{C}\right)^2
\tag{79}
$$

*Proof.* To show (76), (77), (78), we use the expression $\boldsymbol{x}_t - \boldsymbol{x}^\star = P_t(\boldsymbol{A})(\boldsymbol{x}_0 - \boldsymbol{x}^\star)$ (Proposition 2.1) and then evaluate $\|P_t\|_\mu^2 = \int_{\mathbb{C}\setminus\{0\}} |P_t|^2 \, \mathrm{d}\mu$ (Theorem 2.1).

For (76), the value of $\|P_t\|_\mu^2$ follows directly from Theorem 2.3, which states that the value for the optimal residual polynomial $P_t$ is

$$\frac{1}{\sum_{k=0}^t |\phi_k(0)|^2} = \frac{1}{\sum_{k=0}^t \frac{C^{2k}}{K_{k,R}^2}}. \tag{80}$$

A simple proof by induction shows that for the asymptotically optimal algorithm (20), the following expression holds for all $t \geq 0$:

$$\boldsymbol{x}_t - \boldsymbol{x}^\star = \left( \left(\frac{R}{C}\right)^{2t} + \left(1 - \left(\frac{R}{C}\right)^2\right) \sum_{k=1}^t \left(1 - \frac{\boldsymbol{A}}{C}\right)^k \left(\frac{R}{C}\right)^{2(t-k)} \right) (\boldsymbol{x}_0 - \boldsymbol{x}^\star) \tag{81}$$

Thus,

$$\begin{aligned} P_t(\lambda) &= \left(\frac{R}{C}\right)^{2t} + \left(1 - \left(\frac{R}{C}\right)^2\right) \sum_{k=1}^t \left(1 - \frac{\lambda}{C}\right)^k \left(\frac{R}{C}\right)^{2(t-k)} \\ &= \left(\frac{R}{C}\right)^{2t} \phi_0(\lambda) + \left(1 - \left(\frac{R}{C}\right)^2\right) \sum_{k=1}^t \frac{K_{k,R}}{C^k} \phi_k(\lambda) \left(\frac{R}{C}\right)^{2(t-k)}, \end{aligned} \tag{82}$$

which concludes the proof of (77), as

$$\|P_t\|_\mu^2 = \left(1 - \left(\frac{R}{C}\right)^2\right)^2 \sum_{k=1}^t \frac{K_{k,R}^2}{C^{2k}} \left(\frac{R}{C}\right)^{4(t-k)} + \left(\frac{R}{C}\right)^{4t}. \tag{83}$$

By equation (52),

$$\boldsymbol{y}_t - \boldsymbol{x}^\star = \left(1 - \frac{\boldsymbol{A}}{C}\right)^t (\boldsymbol{y}_0 - \boldsymbol{x}^\star) = \frac{K_{t,R}}{C^t} \phi_k(\boldsymbol{A})(\boldsymbol{y}_0 - \boldsymbol{x}^\star) \tag{84}$$

Thus, for the $\boldsymbol{y}_t$ iterates, $\|P_t\|_\mu^2 = \frac{K_{t,R}^2}{C^{2t}}$, and (78) follows.

Now, $\xi_{\text{opt}}(t) \leq \xi_{\text{asymp}}(t), \forall t \geq 0$ is a consequence of $\xi_{\text{opt}}(t)$ being the rate of the optimal algorithm. And

$$\lim_{t\to\infty} \frac{\xi_{\text{opt}}(t)}{\xi_{\text{GD}}(t)} = \lim_{t\to\infty} \frac{\frac{C^{2t}}{K_{t,R}^2}}{\sum_{k=0}^t \frac{C^{2k}}{K_{k,R}^2}} = 1 - \frac{R^2}{C^2} \tag{85}$$

follows from Proposition D.1. To show $\lim_{t\to\infty} \frac{\xi_{\text{opt}}(t)}{\xi_{\text{GD}}(t)} = 1 - \frac{R^2}{C^2}$, which concludes the proof, we rewrite

$$\xi_{\text{asymp}}(t) = \left(\frac{R}{C}\right)^{2t} \left( \left(1 - \left(\frac{R}{C}\right)^2\right)^2 \sum_{k=1}^t \frac{1}{Q_{k,R}} \left(\frac{R}{C}\right)^{2(t-k)} + \left(\frac{R}{C}\right)^{2t} \right), \tag{86}$$

using that by definition, $Q_{k,R} = R^{2k}/K_{k,R}^2$. Now, let $c_\epsilon \in \mathbb{Z}_{\geq 0}$ such that

$$\sum_{k=c_\epsilon}^\infty \left(\frac{R}{C}\right)^{2k} \leq \epsilon. \tag{87}$$

Using the same argument as in Proposition D.1 (see (68)), for $t$ large enough and $k \in [t - c_\epsilon, t]$,

$$\int_k^t \frac{d}{ds} Q_{s,R} \, \mathrm{d}s \leq 2\epsilon Q_{t,R}. \tag{88}$$

Hence, for $t$ large enough,

$$
\left(1 - \left(\frac{R}{C}\right)^2\right)^2 \sum_{k=1}^{t} \frac{1}{Q_{k,R}} \left(\frac{R}{C}\right)^{2(t-k)} + \left(\frac{R}{C}\right)^{2t}
$$

$$
= \left(1 - \left(\frac{R}{C}\right)^2\right)^2 \left( \sum_{k=t-c_\epsilon}^{t} \frac{1}{Q_{t,R} - \int_k^t \frac{d}{ds} Q_{s,R}} \left(\frac{R}{C}\right)^{2(t-k)} + \sum_{k=1}^{t-c_\epsilon} \frac{1}{Q_{k,R}} \left(\frac{R}{C}\right)^{2(t-k)} \right) + \left(\frac{R}{C}\right)^{2t}
$$

$$
\leq \left(1 - \left(\frac{R}{C}\right)^2\right)^2 \left( \frac{1}{(1-2\epsilon)Q_{t,R}} \sum_{k=t-c_\epsilon}^{t} \left(\frac{R}{C}\right)^{2(t-k)} + \sum_{k=1}^{t-c_\epsilon} \left(\frac{R}{C}\right)^{2(t-k)} \right) + \epsilon
$$

$$
\leq \left(1 - \left(\frac{R}{C}\right)^2\right) \left( \frac{1}{(1-2\epsilon)Q_{t,R}} + \left(1 - \left(\frac{R}{C}\right)^2\right)\epsilon \right) + \epsilon,
$$

$$
\tag{89}
$$

which can be made arbitrarily close to $\left(1 - \left(\frac{R}{C}\right)^2\right) \frac{1}{Q_{t,R}}$ by taking $\epsilon > 0$ small enough. Plugging this into (86), we obtain that we can make $\xi_{\text{asymp}}(t)$ arbitrarily close to $\left(1 - \left(\frac{R}{C}\right)^2\right) \left(\frac{R}{C}\right)^{2t} \frac{1}{Q_{t,R}} = \left(1 - \left(\frac{R}{C}\right)^2\right) \xi_{\text{GD}}(t)$ by taking $t$ large enough. $\qquad\square$

