# OpenReview forum: "Average-case Acceleration for Bilinear Games and Normal Matrices"
_ICLR.cc/2021/Conference — ICLR 2021 Poster_

### Official Review · AnonReviewer3 · 2020-10-25
**interesting average-case analysis of first-order methods**

**Rating:** 7
**Confidence:** 2

**Review:**

In this submission, first-order methods for solving smooth games are studied in the average case. In particular, first-order methods are derived and studied that are average-case optimal for certain optimization problems. In particular average-optimal first-order methods for solving zero-sum minimax games are presented. Also for finding the root of non-symmetric affine operators average-case optimal operators are derived if either the relevant matrix is normal or the eigenvalues are supported in a disk. Some experiments with the derived methods are conducted but the focus lies clearly on the theoretical results.

For the analysis of first-order methods a well-known and elegant connection to residual polynomials is used. The analysis is quite technical and rather densely written. I find the line of research natural and in particular I find it interesting that average-case optimal methods (under certain assumptions) can explicitly be derived.

As a non-expert in this area I had some difficulties in following the analysis and I had to read some of the related work to understand the contribution of this submission. My suggestion would be to add maybe one page to the introduction that describes in more detail the context and the relevant terminology.

---

> ### Author Response · Authors · 2020-11-23
> **Answer to Reviewer 3**
>
> We appreciate the comments of the reviewer. We have uploaded an additional pdf which shows the difference between the old submission and the new one, so that the reviewers can quickly check the changes that have been made.
>
> Although we did not make significant changes to the introduction, we did improve the clarity of Subsection 2.4, and to ease the reading we added a more detailed discussion of how our results compare to previous work (see the answer to the last question of Reviewer 4 for more detail). We hope that these changes make it easier to grasp the contribution of our paper. We are aware that to have a thorough comprehension of our paper, one may also have to take a look at Pedregosa & Scieur (2020) and at Azizian et al. (2020), but we find it difficult to avoid.

---

### Official Review · AnonReviewer1 · 2020-10-28
**Reviewer 1**

**Rating:** 7
**Confidence:** 4

**Review:**

The paper combines two recent areas of interest in optimization for machine learning: average-case analysis for acceleration and learning in bilinear games.  Average-case optimal methods are proposed for some distributions on eigenvalues for the Jacobian of the games vector-field. Also, a connection is made between average-case optimal methods in bilinear games and optimally solving the Hamiltonian.


Strengths:

The paper presents methods that improve on worst-case optimal algorithms for learning in games by using average-case optimal algorithms.  Numerical experiments confirm a reasonable benefit in the ill-conditioned regime.

The algorithm in (11) is easy to implement in the asymptotic regime -- ex., with (19).

Theorem 3.1 which connects optimal methods with minimization on the Hamiltonian highlights a recurring theme in the analysis of these kinds of problems, which other authors may be able to use in the future.


Weaknesses:

There are limited experimental results,  but they do succinctly show the expected results from average-case optimization in their setup.

It’s not clear how to apply these methods towards more realistic problems we may want to solve.  For example, in optimizing GANs, how do the distribution of the eigenvalues of the Jacobian compare during training (for medium-scale problems with say 1-5k parameters) with the distributions you investigate?  Is there a useful distinction between how one would implement worst-case and average-case algorithms in situations where we don’t have a tight understanding of the eigenvalue distribution?

It’s not clear to me what assumptions are needed at some points in the paper.  When do we need A to be skew-symmetric (i.e., a zero-sum bilinear game)? Should Thm 3.1 be for “zero-sum bilinear games” instead of bilinear games in general?  If the zero-sum assumption is required, it might be worthwhile emphasizing when it is/isn’t needed when talking about “bilinear games”.  Prop 5.2 says it requires the assumptions of Theorem 4.2 which again says it requires the assumptions of Theorem 2.1.  I find this difficult to read and would appreciate a summary of the assumptions required directly at each step.


My recommendation is to accept the paper with a 7.  Studying learning in games is an important problem in machine learning, and establishing average-case algorithms complements recent worst in worst-case bounds for bilinear games. The asymptotic algorithms presented are simple to use for some eigenvalue distributions. The experimental results are weak, but almost unnecessary given the strong theoretical results.  If there were proposed ways (and results) to leverage the average-case algorithm variants in more-general setups like GANs I would raise the score.


The following point did not affect my score but may help the authors:

If you are pressed for space to add details, it’s not clear to me that you need to re-state the result Prop 5.1 in the main body.

It’s unclear to me what is meant by “For bilinear games, since the average-case optimal algorithm is average-case optimal algorithm of an optimization algorithm.”

Some terms are not defined explicitly in the main body of the text before being used.  For example, D_{C, R}. However, this did not strongly hinder comprehension.

The paper states “This result complements [1], proved that Polyak Heavy Ball algorithm on the Hamiltonian is asymptotically worst-case optimal.”  It’s unclear to me how this is shown in their paper. Could you elaborate on this?

Wa¨ıss Azizian, Damien Scieur, Ioannis Mitliagkas, Simon Lacoste-Julien, and Gauthier Gidel. Accelerating smooth games by manipulating spectral shapes, 2020.

---

> ### Author Response · Authors · 2020-11-23
> **Answer to Reviewer 1**
>
> We thank the reviewer for their thorough and helpful review. The major concern raised by the reviewer is the application of our method to more realistic settings, such as GANs. In the response below; we give some insights and directions to apply our method in such settings. We have uploaded an additional pdf which shows the difference between the old submission and the new one, so that the reviewers can quickly check the changes that have been made.
>
> **Q**: “It’s not clear how to apply these methods towards more realistic problems we may want to solve. For example, in optimizing GANs, how do the distribution of the eigenvalues of the Jacobian compare during training (for medium-scale problems with say 1-5k parameters) with the distributions you investigate?”
>
> **A**: We have now added to the paper Figure 1 showing the spectrum of a GAN at initialization. This shows a spectral density with strong similarities to the circle law, i.e., it satisfies Assumption 5. However, there are still technical limitations to the direct implementation of the proposed method to GANs. For example, this work is limited to deterministic methods, although we hope to open the way to more efficient stochastic methods in future work.
>
>
> **Q**: “Is there a useful distinction between how one would implement worst-case and average-case algorithms in situations where we don’t have a tight understanding of the eigenvalue distribution?”
>
> **A**: When one doesn't have a tight understanding of the eigenvalue distribution it is possible to approximate it with a parametric model: for instance, [Martin et al., Implicit Self-Regularization in Deep Neural Networks: Evidence from Random Matrix Theory and Implications for Learning](https://arxiv.org/abs/1810.01075) show in Section 4 that the spectrum of neural networks is surprisingly predictable in the over-parametrized case. We expect such property to hold for GANs (in Figure 1, we intuitively see that the GAN spectrum has a circularly symmetric distribution that could easily be fitted), although this is out of the scope of our study.
> Finally, in the case where one uses an optimal average-case algorithm on a problem whose eigenvalues distribution does not fit the expected one, it may happen that the average-case method will be suboptimal (compared to the worst-case one) for early iterations. Hopefully, as we showed in Section 5 of our paper, the optimal algorithm in the average-case (and its rate) converges to the optimal algorithm in the worst-case.
>
> **Q**: “It’s not clear to me what assumptions are needed at some points in the paper. [...] Prop 5.2 says it requires the assumptions of Theorem 4.2 which again says it requires the assumptions of Theorem 2.1.”.
>
> **A**: $A$ is skew-symmetric in the part relating to bilinear games, that is, Section 3 and Proposition 5.1 in Section 5. Throughout the paper, when we talk about bilinear games we refer to _zero-sum_ bilinear games, and we drop the _zero-sum_ part for shortness and because it is also the terminology used by Azizian et al. We already mentioned that we deal with zero-sum games in the first sentence of Section 3, but we have emphasized it by stating it explicitly. In Theorems 4.1 and 4.2 we no longer state the results referring to the Assumptions of Theorem 2.1, but we refer to the Assumptions 1 and 2 directly. In Proposition 5.1 we now state that the setting is the one of Theorem 3.1 and in Proposition 5.2 we keep referring to the assumptions of Theorem 4.2 for shortness.
>
> **Q**: “It’s unclear to me what is meant by “For bilinear games, since the average-case optimal algorithm is average-case optimal algorithm of an optimization algorithm.”.
>
> **A**: We have rephrased the sentence, removing the typo. What we mean is that in Theorem 3.1 we see that the average-case optimal first-order method to find solutions of a *bilinear game* is the average-case optimal first-order method for the *minimization of the Hamiltonian* of the game. Hence, we just need to find the asymptotic version of the average-case optimal first-order method for the minimization of the Hamiltonian of the game, which was studied by Scieur & Pedregosa (2020).
>
> **Q**: “Some terms are not defined explicitly in the main body of the text before being used. For example, D_{C, R}.”.
>
> **A**: We now define this instance the first time it is used.
>
> **Q**: “The paper states “This result complements [1], proved that Polyak Heavy Ball algorithm on the Hamiltonian is asymptotically worst-case optimal.” [...] Could you elaborate on this?”.
>
> **A**: In section 3, we added a paragraph comparing our average-case results with those of Azizian et al. (2020). We also added a pointer to section 3 in the introduction after this sentence.

---

### Official Review · AnonReviewer4 · 2020-10-29
**Extend previous average-case complexity work from symmetric to normal matrices (i.e., diagonalizable)**

**Rating:** 6
**Confidence:** 3

**Review:**

This paper considers the average-case complexity analysis for bilinear game and normal operator problem.
Instead of worst-case analysis, they measure the expected complexity with respect to the random inputs, i.e., matrix $A$, points $x^*$ and $x_0$.
To do this, they first show that this expectation is related to an integral of polynomials associated with the first-order method (Theorem 2.1). Then they invoke some previous work to indicate how to minimize the integral.

Although the authors pointed out the difference between this paper and the previous work Pedregosa & Scieur (2020) in Section 2.4, the contribution of this paper is still not very informative to me.
They said that Pedregosa & Scieur (2020) considered the symmetric matrix $A$, while this work extends to normal (diagonalizable) matrix $A$, i.e., $AA^T=A^TA$ (Assumption 2). So what’s the difficulty for extending symmetric matrix to diagonalizable matrix? It seems that the analysis only needs the spectral information of the matrix.
For the other difference, the domain of the integral is real or complex. So what’s the benefit of introducing complex instead of real.

Other comments: The writing can be improved by e.g. providing more discussions instead of just stacking the propositions and theorems.

---

> ### Author Response · Authors · 2020-11-23
> **Answer to Reviewer 4**
>
> The reviewer raises some very valid points regarding the connection of our work with Pedregosa & Scieur (2020). We hope that our answer and the changes to the manuscript will influence the reviewer's opinion about the novelty with respect to related work. We have uploaded supplementary material showing the difference between the old submission and the new one, so that the reviewers can quickly check the changes that have been made.
>
> **Q**: “[...] the contribution of this paper is still not very informative to me [...] So what’s the difficulty for extending symmetric matrix to diagonalizable matrix?”
>
> **A**: There are crucial differences from the symmetric to the normal setting. The major technical difficulty arises at Theorem 2.1 (analog of Theorem 2.1 in Pedregosa and Scieur 2020, but in the complex plane): indeed, the eigenvalues of normal matrices are potentially complex. This makes the problem of finding the optimal polynomial much more difficult. Indeed:
> - In the real case (i.e. symmetric matrices), the sequence of optimal polynomials is the sequence of orthogonal polynomials w.r.t. the weight function lambda mu(lambda).
> - In our case, the optimal polynomial no longer follows such a nice property, as the optimal one is a linear combination of orthogonal polynomials (see Theorem 2.3). This also implies that the optimal algorithm in the average case for normal matrices may not follow a three-term recurrence (as opposed to the ones in Pedregosa and Scieur, 2020).
>
> **Q**: “So what’s the benefit of introducing complex instead of real.”
>
> **A** : Complex eigenvalues allow us to analyze the average-case convergence of non-symmetric vector fields like the ones that arise in GAN training. For example, the eigenvalues of the matrix in bilinear games lie in the imaginary line. Therefore, their expected spectral density is a measure in the complex plane and it would not be possible to analyze them using real eigenvalues as has been done previously in (Pedregosa and Scieur, 2020).
>
> **Q**: “The writing can be improved by e.g. providing more discussions instead of just stacking the propositions and theorems.”
>
> **A**: We have uploaded an additional pdf which shows the difference between the old submission and the new one. We have improved the writing mainly in the following ways:
> - In Subsection 2.4 we provide a clearer explanation of the limitations we encounter when going from the symmetric matrix setting to the normal matrix setting.
> - In Section 3 we have added an in-depth comparison paragraph with previous work, and in Subsection 4.1 we have emphasized the comparison with previous work by adding a paragraph title.
> - In the Conclusion, we have added a paragraph about how our framework could possibly be used for GAN training.

---

### Decision · Program_Chairs · 2021-01-07
**Final Decision**

**Decision:**

Accept (Poster)

**Comment:**

The reviewers overall liked the paper and the mathematical contribution seems substantial and elegant. The two dominant concerns were whether this is really applicable to GANS, and whether the increment from symmetric to normal matrices (e.g., real to complex eigenvalues, still unitary eigenvectors) was significant enough. Our consensus is that this result is a step toward analyzing practical GANS, and (based on the authors' response) that the extension to complex eigenvalues was substantial enough. Hence I'm happy to recommend the paper.